# Multi-Spectral Image Classification Based on an Object-Based Active Learning Approach

**Tengfei Su** **, Shengwei Zhang \* and Tingxi Liu**

College of Water Conservancy and Civil Engineering, Inner Mongolia Agricultural University, Hohhot 010018, China; stf1987@imau.edu.cn (T.S.); txliu1966@imau.edu.cn (T.L.)

\* Correspondence: zsw@imau.edu.cn; Tel.: +86-186-8606-0015

**Abstract:** In remote sensing, active learning (AL) is considered to be an effective solution to the problem of producing sufficient classification accuracy with a limited number of training samples. Though this field has been extensively studied, most papers exist in the pixel-based paradigm. In object-based image analysis (OBIA), AL has been comparatively less studied. This paper aims to propose a new AL method for selecting object-based samples. The proposed AL method solves the problem of how to identify the most informative segment-samples so that classification performance can be optimized. The advantage of this algorithm is that informativeness can be estimated by using various object-based features. The new approach has three key steps. First, a series of one-against-one binary random forest (RF) classifiers are initialized by using a small initial training set. This strategy allows for the estimation of the classification uncertainty in great detail. Second, each tested sample is processed by using the binary RFs, and a classification uncertainty value that can reflect informativeness is derived. Third, the samples with high uncertainty values are selected and then labeled by a supervisor. They are subsequently added into the training set, based on which the binary RFs are re-trained for the next iteration. The whole procedure is iterated until a stopping criterion is met. To validate the proposed method, three pairs of multi-spectral remote sensing images with different landscape patterns were used in this experiment. The results indicate that the proposed method can outperform other state-of-the-art AL methods. To be more specific, the highest overall accuracies for the three datasets were all obtained by using the proposed AL method, and the values were 88.32%, 85.77%, and 93.12% for "T1," "T2," and "T3," respectively. Furthermore, since object-based features have a serious impact on the performance of AL, eight combinations of four feature types are investigated. The results show that the best feature combination is different for the three datasets due to the variation of the feature separability.

**Keywords:** object-based image analysis; active learning; random forest; feature category

## 1. Introduction

The number of remote sensing platforms is continually increasing, and they are producing a tremendous amount of earth-observation image data. However, it is challenging to extract essential information from these images [1–3]. Image classification plays a fundamental role in this task, and there have been great efforts dedicated to the development of image classification methods, such as data mining, deep learning [4,5], and object-based image analysis (OBIA). Among these techniques, OBIA has been widely applied to high spatial resolution image classification [6–8] because it can take high advantage of the spatial information that is captured within these images. Besides, many researchers consider OBIA to be an interesting and evolving paradigm for various applications, e.g., agricultural mapping [9,10], forest management [11,12], and urban monitoring [13,14].

Recent studies have frequently reported that OBIA can achieve good performance in remote sensing applications [7,15–17], mainly due to two reasons. For one thing, OBIA-based classification algorithms can reduce or even eliminate salt-and-pepper noises, which often exist in the classification results of pixel-based strategies [18]. This is because image segmentation is generally the first step in OBIA. This procedure partitions an image into several non-overlapping and homogeneous parcels (or segments, objects) so that noisy pixels do not cause errors in classification results [19]. Lv et al. [20] developed an object-based filter technique that can reduce image noise and accordingly improve classification performance. Additionally, compared with pixel-based classification approaches, OBIA is capable of utilizing various image features. This is because the processing unit in OBIA is an object instead of the pixel, so this paradigm can effectively employ object-level information [21,22]. At the object level, it is convenient to extract geometric and spatial contextual features which may enhance the discriminative power of the feature space. Driven by Tobler's first law of geography, Lv et al. [23] extracted object-based spatial–spectral features to enhance the classification accuracy of aerial imagery.

Though OBIA has the mentioned above merits, its potential has not yet been fully utilized. The first challenge originates from the step of image segmentation. When segmenting a remote sensing image, over- or under-segmentation errors often occur, even if a state-of-the-art segmentation algorithm is employed [24–26]. OBIA classification suffers greatly from segmentation errors, especially in the case of under-segmentation [27,28]. The second issue is that, similarly to traditional pixel-based approaches, the performance of OBIA is limited by the quantity and quality of its training samples. For a number of real applications, it is required that training samples should be as few as possible to achieve sufficient classification accuracy because in some situations, collecting samples is costly. Examples include mapping tasks in hazardous or remote areas, such as post-earthquake cities, landslide affected zones, and outlying agricultural fields [29–32]. The accessibility of these areas is often limited, so sample acquisition is difficult or impossible by human on-site visits. Sending drones or purchasing satellite images of a much higher spatial resolution may help obtain the ground truth information, but the cost would be raised significantly. Accordingly, the real problem is how to collect the most useful samples within a limited sample-collection budget. Active learning (AL) aims to provide a solution to this issue by guiding the user to select samples that may optimally increase classification accuracy [33–35]. In this way, users do not have to spend time or cost on attempting to get the information of some useless samples, which are either redundant or produce little effects on improving classification performance. Under the guidance of an AL method, a relatively small sample set that is capable of optimizing the separability of a classifier can be achieved, fulfilling the objective of getting a sufficiently high classification accuracy with a limited number of training samples. Though this seems tempting, it is challenging to apply AL techniques to OBIA, since related studies are comparatively rare.

In the field of remote sensing, it is common to apply AL to pixel-based classification. In these studies, AL deals with supervised classification problems when a small set of training pixels is available. By iteratively adding new samples into the training set, AL may help raise classification performance. Thus, it is clear that the key objective of AL is to identify the most informative samples that optimally improve classification accuracy. In an implementation, an AL method aims to identify the sample with the largest classification uncertainty [36–38], and there are three main ways to achieve it [39]. The first strategy uses information gain that is generally formulated by using Shannon entropy. The name of the second category is breaking-tie (BT), and it adopts the criterion of posterior probabilistic difference. The third one's name is margin sampling, and it mostly combines with a support vector machine. There are some examples, which are introduced as follows. Tuia et al. [34] constructed a BT algorithm to enable AL to detect new classes. By fusing entropy and BT approaches, Li et al. [40] developed a Bayesian-based AL algorithm for hyperspectral image classification. Inspired by the idea of region-partition diversity, Huo and Tang [41] implemented a margin sampling-based AL method. Xu et al. [39] proposed a patch-based AL algorithm by considering the BT criterion. Sun et al. [42] designed three AL methods based on a Gaussian process classifier and three modified BT strategies.

Compared to the above-mentioned pixel-based AL methods, there have been relatively few efforts to document object-based AL, but there are a few examples. Liu et al. [43] proposed an AL scheme based on information gain and the BT criterion to classify PolSAR imagery. Based on margin sampling and multiclass level uncertainty, Xu et al. [44] implemented an object-based AL strategy for earthquake damage mapping. Ma et al. [45] developed an object-based AL approach by considering samples with zero and large uncertainty. These studies have made conspicuous contributions to OBIA and AL, but some issues still exist. This article focuses on two of them.

The first issue is that when it comes to OBIA, feature space is generally much more complicated than the counterpart of the pixel-based analysis. In the pixel-based paradigm, one generally calculates features based on a single pixel, or a window of pixels centered at the target pixel; therefore, this process can only capture the information of a limited spatial range. In comparison, object-based features contain more information, such as geometric and spatial contextual cues. The more complicated feature space brings about a new challenge to the traditional AL algorithms because AL relies on feature variables to quantify uncertainty. Accordingly, the more complicated feature set in OBIA requires new AL methods, a fact which motivated this work. For this purpose, this paper presents a new strategy for uncertainty measurement based on one-against-one (OAO) binary random forest (RF) classifiers. Though RF is a popular and successful classifier in remote sensing [46] and OBIA [47–49], it is interesting to test the OAO binary RF for AL method construction and to see if it can fine estimate uncertainty when using object-level features.

Secondly, to the best of our knowledge, none of the previous studies investigated the effects of object feature types on AL performance. Though some previous studies have explored how to determine the most discriminative features, most of them have focused on hyperspectral image classification [50,51]. There have been even fewer similar studies on OBIA. OBIA generally provides four types of object-based features, including geometric, spectral, textural, and contextual feature categories [49,52]. The discriminative power of the four feature types may vary wildly in different scenarios, and this can produce a great influence upon the AL process. However, according to previous works, it is unclear how classification performance behaves when an object-based AL uses different combinations of the four feature categories. In this work, the evaluation part considers the effects of different object-feature types, and we consider this as a contribution in terms of experimental design.

According to the issues described above, this article proposes a new object-based AL algorithm by using an OAO-based binary RF model. It combines the posterior probabilistic outputs with a modified BT criterion to quantify classification uncertainty in a detailed way. Additionally, with the proposed AL approach, different combinations of the four object feature categories are tested to see which combination is the most appropriate for object-based AL.

It is as follows to organize this paper. Section 2 details the principle of the new object-based AL. Section 3 shows the experimental results, as well as tests on the effects of different combinations of the four object feature types on AL performance. Additionally, there are comparisons between the proposed algorithm and other competitive AL approaches. Sections 4 and 5 provide discussion and conclusion, respectively.

## 2. Methodology

In this part, we first introduce the basic concepts of AL in Section 2.1, for the convenience of describing the proposed algorithm. Then, Section 2.2 discusses the implementation of an object-based AL, followed by a detailed description of the proposed AL approach in Sections 2.3 and 2.4. The last sub-section introduces the object-based features used in this study.

### 2.1. Basics of Active Learning

An AL method consists of 5 parts, including a training set $T$, a classifier $C$, a pool of unlabeled samples $U$, a query function $Q$, and a supervisor $S$. Table 1 describes a simple AL process. Step 2 and 3 make up an iterative process in which the most important component should be $Q$ since it determines

whether the process can select good samples. Generally speaking, for successful AL, the classification accuracy of the output $T$ should be evidently higher than that produced by using the initial $T$. This depends on whether the samples found by $Q$ are beneficial to the classification performance of $C$. To enhance the readability of this article, the meanings of the abbreviations and letter symbols related to the principle of an AL approach are listed in Table A1 of the Appendix A section.

**Table 1.** The steps of a simple active learning (AL) algorithm.

| **Input:** $T$, $C$, $U$ |
| --- |
| **Output:** an enlarged $T$ |
| 1. Train $C$ by using $T$; initialize $Q$ by using $C$; |
| 2. Find sample(s) in $U$ by using $Q$; let $S$ provide label information for the sample(s); add the sample(s) into $T$; remove the sample(s) from $U$; |
| 3. Update $Q$ by using $C$ and $T$; Go to step 2 if the updated $T$ meets the stopping criterion; |
| 4. Output $T$. |

Intuitively, a good $Q$ can identify samples with the highest classification uncertainty, because many hold that the uncertain samples can help raise the discriminative power of $C$. Accordingly, AL related studies have all focused on the design of $Q$. In implementation, $Q$ is a criterion that is used to measure the classification uncertainty of unlabeled samples.

*2.2. Object-Based Active Learning*

The work-flow of an object-based AL is similar to that delineated in Table 1. However, since the processing unit in OBIA is an object, a sample differs from that of pixel-based methods. For pixel-based AL, a sample consists of a pixel label and a feature vector of that pixel. For object-based AL, a sample corresponds to an object, and so does its label and feature vector. This directly results in 2 differences between the 2 AL types.

First, the searching space of object-based AL can be much smaller than the pixel-based counterpart, because, for the same image, the number of objects is much lower than that of pixels. This tends to simplify the sample searching process of $Q$, but it may not be the case, mainly due to the next aspect. The second difference resides in the feature vector contents because, in OBIA, there are more feature types for a processing unit. OBIA allows for the extraction of geometric information and statistical features (e.g., mean, median, and standard deviation values for spectral channels). Thus, the object feature space can be bigger and more complicated, bringing a great challenge to the computation of $Q$.

Accordingly, $Q$ in object-based AL should be able to finely estimate the appropriateness of an unlabeled sample. Previous studies concerning this aspect have attempted to split a multiclass classification problem into a set of binary classification procedures so that each class can be treated carefully in the sub-problem. To do so, there are mainly 2 schemes: one-against-all (OAA) and one-against-one (OAO) [42,53]. Suppose that there are $L$ classes to be classified in an image; then, OAA divides the $L$-class problem into $L$ binary classification cases. The user trains each of these binary classifiers by using 2 groups of samples, including the samples of one class and the samples of the other $L - 1$ classes. Though OAA is a widespread scheme, it may suffer from imbalanced training due to the allocation of the samples of $L - 1$ classes into one training set. In comparison, OAO can avoid this issue, but it has to construct $L \cdot (L - 1)/2$ binary classifiers, which is a little more complicated than the OAA approach. This work adopts the OAO strategy due to the above-mentioned merit.

*2.3. Random Forest-Based Query Model*

Breiman proposed random forest (RF) [54], and during recent years, it has been successfully applied to diverse remote sensing applications. As indicated by its name, the most intriguing feature of RF is its randomness embodied in 2 aspects. First, RF is composed of a large set of decision trees (DT), each of which is trained by using a sample subset that is randomly selected from the total

training set. This procedure adopts a bootstrap sampling method that can enhance the generalizability and robustness for RF. Second, each DT exploits a subset of feature variables, which can help avoid over-fitting and further improve robustness.

This work proposes a binary RF-based query model and applies it to object-based AL. The key component of this query model is to quantify the appropriateness of the tested samples, and then, the model selects the most appropriate sample(s). To achieve this, we designed 3 steps in the proposed algorithm.

Step (1): initialization. According to the OAO rule, for $L$ ($L > 2$) classes, $L \cdot (L - 1)/2$ binary classifiers are built up by using the initial training sample set. In implementation, each binary RF is trained by using the samples of only 2 classes.

Step (2): test sample processing. A test sample is classified by using the initialized binary classifiers. Each of them can produce a label ($l$) and the associated probability value ($p$). In this way, $L \cdot (L - 1)/2$ pairs of $l$ and $p$ can be obtained for a test sample.

Step (3): appropriateness estimation. Among the results obtained in the last step, the dominant class can be identified. If the label of this class is $l_d$, then there are $n_d$ binary classifiers that assign $l_d$ to the test sample. It is easy to understand that the maximum value of $n_d$ is $L - 1$, because among the $L \cdot (L - 1)/2$ binary classifiers, there are $L - 1$ classifiers involved with each class. For the $n_d$ classifiers (suppose they make up a set $\mathbf{F}_d$), the one producing the maximum uncertainty is chosen to reflect the degree of appropriateness for the test sample. This process can be formulated by using Equation (1),

$$m_a = \min_{i \in \mathbf{F}_d}(p_{i,1} - p_{i,2}) \tag{1}$$

where $p_{i,1}$ and $p_{i,2}$ represent the probability values of the $i$th binary classifier in $\mathbf{F}_d$; for convenience, it can be prescribed that $p_{i,1} \geq p_{i,2}$; $m_a$ represents the appropriateness measure for a test sample. This equation implicates that the class which is the most confusing with $l_d$ yields the highest level of classification uncertainty. Given that $p_{i,2} = 1 - p_{i,1}$ in the case of binary classification, Equation (1) can be rewritten as

$$m_a = \min_{i \in \mathbf{F}_d}(2\Delta p_{i,1} - 1) \tag{2}$$

which is equivalent to

$$m_a = \min_{i \in \mathbf{F}_d}(p_{i,1} - 0.5) \tag{3}$$

In implementation, Equation (3) is adopted.

Equation (3) is similar to that proposed by Sun et al. [42], except that this work derives $l$ and $p$ by using RF, while Sun's method uses the Gaussian process classifier. In this study, the RF model that is implemented in OpenCV was adopted, and this implementation allowed for the derivation of $p$ only in the context of binary classification. In more detail, $p$ is estimated here by using the ratio between the number of DTs producing one class label and the total number of DTs.

To better understand the proposed query model, Figure 1 illustrates an example. The number of classes ($L$) is 4 so that there are 6 binary classifiers that are constructed by using the initial training set, in which there are 2 samples for each class. For an unlabeled sample $u_i$, each of the 6 classifiers produces a label and an associated probability value. It can be seen that 4 is the dominant class, 2 is the most confusing with 4, and the uncertainty value can be calculated as 0.05 according to Equation (3).

In some real cases, it may occur that more than one dominant class exists after step (2). For example, if the label predicted by classifier $F(2,4)$ is 2 instead of 4, the classes of 1, 2, and 4 have the same number of prediction results, and the three classes can all be considered as the dominant class. The model of Equation (3) cannot handle this situation. To solve this problem, it is defined that among the results of the multiple dominant classes, the minimum value of $p - 0.5$ is used as $m_a$.

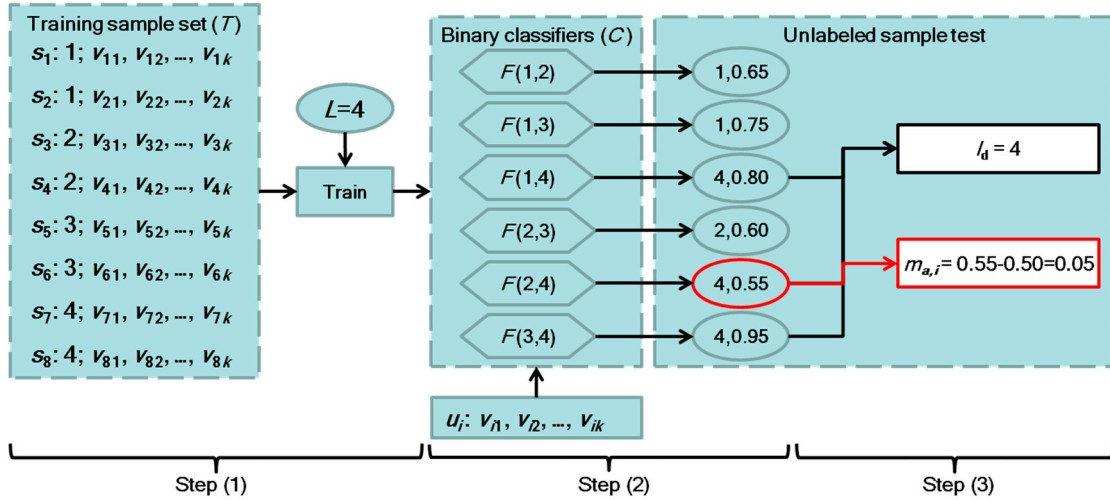

**Figure 1.** An example to illustrate how the proposed query function works.

## 2.4. The Proposed AL Algorithm

### 2.4.1. Details of the Proposed AL

With the query model described in the last sub-section, we can now provide the overall workflow of the proposed AL approach. The red-solid-line box in Figure 2 illustrates the detailed process of the proposed AL method. The most important part is query function ($Q$), which is an adaptation of the 3 steps that are delineated in Section 2.3 because, in the framework of AL, samples of high-level appropriateness should be selected and used to iteratively update $T$, $U$, and $C$. For illustrative purposes, the arrows are numbered to indicate the order of the steps.

It is worth noting that the sorting procedure of $Q$ arranges the tested samples in ascending order because the one with the lowest $m_a$ value is considered to contain the highest uncertainty. What is more, to enable batch mode AL, it is defined that the first $q$ ($q \geq 1$) sorted samples is/are selected in the sample selection step. According to previous research, the batch mode can speed up the calculation efficiency of AL, but it may compromise AL performance. Thus, $q$ is deemed as an important parameter and was analyzed in the experiment.

After the steps of $Q$, $q$ unlabeled samples $\{u_i \mid i = 1, \ldots, q\}$ are selected and labeled by a supervisor/user, and then $U$ and $T$ are accordingly updated: $T = T \cup \{u_i\}$, $U = U \backslash \{u_i\}$. Note that if the number of the remaining samples in $U$ is less than $q$, the whole process is terminated. Another termination condition is that, if the total number of samples in $T$ is greater than a predefined threshold, the AL method ends. The output of the AL algorithm is an enlarged $T$, in which the added samples are expected to raise classification performance.

### 2.4.2. Details of the Whole Processing Chain

The overall process is shown in the upper part of Figure 2. The first step is image segmentation, the objective of which is to partition an image into several non-overlapping and homogeneous segments. In OBIA, unsupervised segmentation algorithms are generally used, and this study also follows this road. A frequently adopted method, called multi-resolution segmentation (MRS) [55], was used in this work. MRS is an unsupervised region merging technique. Initially, it treats each pixel as a single segment. Then, according to a heterogeneity change criterion based on spectral and geometric metrics [55], an iterative region merging process is initiated. During this process, only the segments that are mutually best fitting are merged. The mutual best fitting rule is explained in [55], and it can effectively reduce inappropriate merging. MRS has 3 parameters, including a shape parameter, a compactness parameter, and scale. The former two are both within the range of (0,1) and serve as weights in spectral and geometrical heterogeneity measures [55]. Scale is generally considered as the

most important parameter because it controls the average size of the resulted segmentation. A high scale leads to large segments, and thus under-segmentation errors tend to occur, while a low scale results in small segments; hence, over-segmentation errors may be produced. To avoid this issue, the optimal scale has to be exploited. Section 3.2 provides the related details on this problem.

After segmentation, samples are then prepared for subsequent steps. As shown in Figure 2, there are training ($T$) and unlabeled ($U$) samples. The former contains 2 parts: (1) the class label and (2) the feature vector. The latter only has part (2). Thus, feature vector should be extracted for all of the samples in $T$ and $U$. Since the segment is the processing unit, segment-level features are computed, the details of which are given in Section 2.5.

The next procedure is the proposed AL, which is described in the aforementioned sub-sections. The output of AL is an enlarged $T$, acting as the final training set for a classifier. In this study, an RF classifier is applied. Note that this RF is different from those mentioned in Section 2.3, since this RF is a standard multi-class classifier, while those used in the AL query model are binary classifiers and are used for uncertainty quantification.

Then, classification can be achieved by using the aforementioned standard RF, which takes the feature vector of each segment as input and predicts a class label for that segment. This RF is trained by using 5-fold cross-validation, and throughout this work, its 2 parameters (the number of decision trees ($N_{\text{tree}}$) and the number of split variables ($m_{\text{try}}$)) are set as 300 and the square root of the total number of features, respectively. This setting was tested to be sufficient for this study.

To output the final classification result, the pixels of a segment are rendered with the same label as the prediction result for that segment. The result can then be used for classification evaluation and illustration.

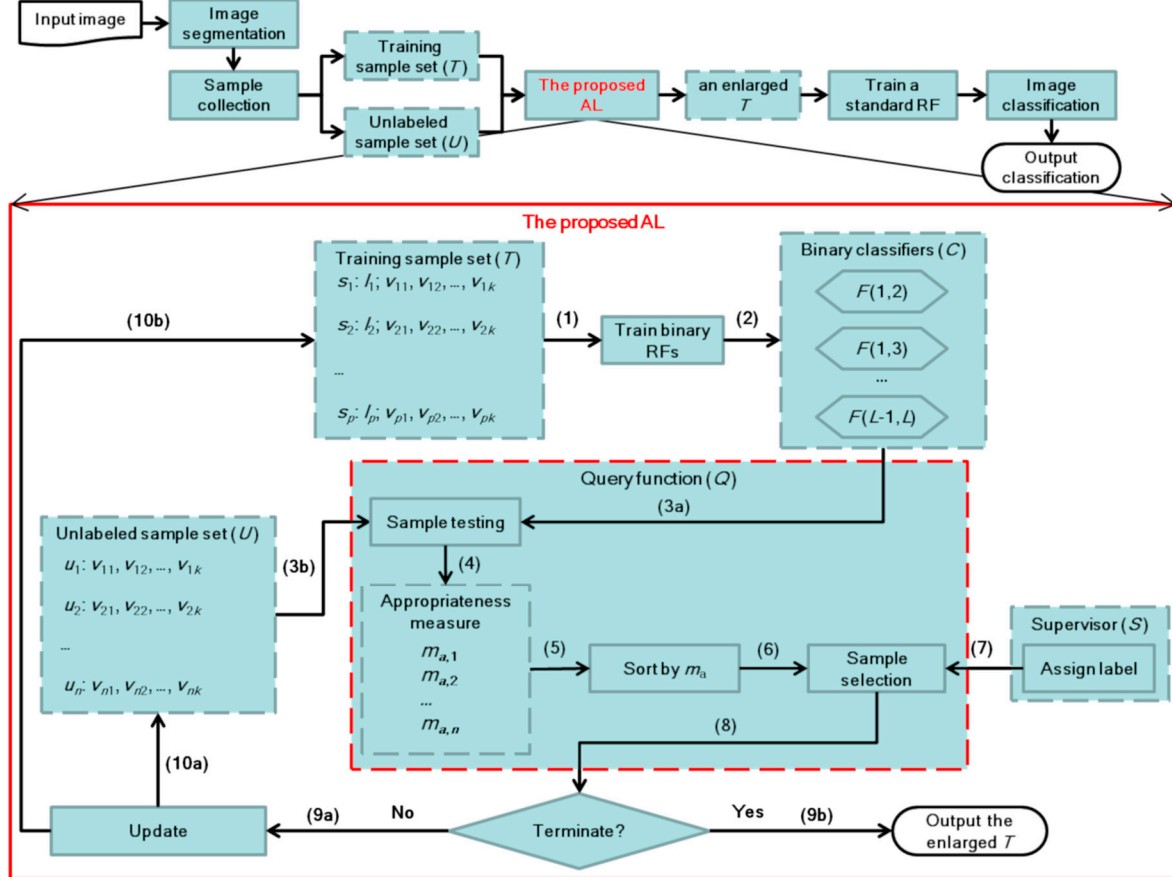

**Figure 2.** Overall workflow of the proposed AL algorithm. The arrows in the red-solid-line box of "the proposed AL" are numbered to indicate the order of the AL process. Note that the letter after the arrow number, e.g., (3a), means the part (a) of step 3.

### 2.5. Object-Based Feature Extraction

In OBIA, a processing unit is represented by several segment-level features that may contain spatial, spectral, textural, and contextual information. This may significantly lengthen the feature vector and complicate the feature space, and it is inclined to produce some influences on AL. To investigate these effects, object-based features that have been frequently applied in OBIA studies are listed here and were tested in the experiment.

There were 4 types of object-based features used in this work, including 10 geometric features, $3 \times B_S$ spectral features, $3 \cdot B_T$ textural features, and $3 \cdot B_S$ contextual features. $B_S$ means the number of spectral channels of the input image, and $B_T$ represents the number of textural feature bands. Table 2 details the information on these features.

In the description of geometric features, the outer bounding box refers to a rectangle bounding the object. The edge direction of such a rectangle is parallel to the edge of the image. Thus, such a bounding box is generally not the minimum one for the object, but this feature is frequently used because it is simple to compute and can reflect the relative geometric direction of an object.

To extract textural features, a grey-level co-occurrence matrix (GLCM) and principal component analysis (PCA) is adopted. At first, 8 GLCM-based textural feature descriptors are calculated for each spectral band. These descriptors include mean, variance, homogeneity, contrast, dissimilarity, entropy, second moment, and correlation [56]. The grey-scale quantification level is set as 32, and 3 processing window sizes ($3 \times 3$, $5 \times 5$, and $7 \times 7$) are utilized to capture multi-scale texture. The co-occurrence shifts in horizontal and vertical directions are both set as 1. This configuration leads to $3 \times 8 \times B_S$ textural feature bands that contain too much redundant information. Accordingly, PCA is used to generate a concise set of textural feature bands. The PCA-transformed feature bands that correspond to the first four principal components are selected to derive object-based textural features. For the datasets used in this study, the details of the PCA results are provided in Section 3.1.

**Table 2.** Object-based features adopted in this study. For spectral, textural, and contextual features, each line of this table corresponds to a feature extracted from one feature band.

| Feature Type | Feature Name | Description |
| --- | --- | --- |
| Geometric | Area ($A$) | $A$ is measured by using the number of pixels. |
| | Perimeter ($P$) | $P$ is calculated by counting the number of edge pixels. |
| | Roundness ($R_o$) | $R_o = P^2/A$ |
| | Rectangular degree ($R_{rec}$) | $R_{rec} = A/A_b$. $A_b$ means the area of the outer bounding box of the object. |
| | length/width ratio ($R_{lw}$) | Ratio of the length and width of the outer bounding box. |
| | Shape index ($I_s$) | $I_s = P/(4 \cdot P_s)$. $P_s$ means the perimeter of a square that has the same area with the object. |
| | Border index ($I_b$) | $I_b = 0.5 \cdot P/(w + l)$. $w$ and $l$ symbolize the width and length of the object, respectively. |
| | Asymmetry ($R_a$) | $R_a$ is defined by comparing an approximated ellipse with the object. Variances in pixel coordinate are used to compute this feature. Readers are referred to the reference book of eCognition for computation details [57]. |
| | Main/secondary direction width ratio ($R_{ms}$) | Ratio of the object widths in main and secondary directions. It is calculated by using the ratio of the two eigenvalues of the covariance matrix of pixel coordinates. |
| | Density ($R_d$) | $R_d = P_s/(1 + (V_x + V_y)^{0.5})$. $P_s$ is similarly defined as in the description of $I_s$. $V_x$ and $V_y$ are the coordinate variance in horizontal and vertical directions, respectively. |

**Table 2.** *Cont.*

| Feature Type | Feature Name | Description |
|:---:|:---:|:---:|
| Spectral | Average value ($S_a$) | The average pixel value for a spectral channel. |
| | Median value ($S_m$) | The median pixel value for a spectral channel. |
| | Standard deviation (STD) ($S_s$) | The standard deviation of pixel value for a spectral channel. |
| Textural | Average value ($T_a$) | Similarly defined as the spectral features, but textural feature bands are used. |
| | Median value ($T_m$) | |
| | STD ($T_s$) | |
| Contextual | Average of contrast ($C_a$) | The mean difference between $S_a$ of the object and the $S_a$s of its neighboring objects. |
| | Median of contrast ($C_m$) | The mean difference between $S_m$ of the object and the $S_m$s of its neighboring objects. |
| | STD of contrast ($C_s$) | The mean difference between $S_s$ of the object and the $S_s$s of its neighboring objects. |

For both spectral and contextual features, average, median, and standard deviation are utilized. The three variables can reflect the statistical pattern for an object.

Among the 4 types of object-based features, the spectral feature is the most frequently used. Textural and contextual features are extracted based on spectral features, so the 3 types may contain some dependence. On the other hand, geometric features are independent of the other 3 types, but whether positive effects on classification performance are produced is dependent on the application at hand. In the following experiment, different combinations of the 4 feature types were tested to investigate their influences on AL performance.

## 3. Dataset

### 3.1. Satellite Image Data

Three sets of high spatial resolution multispectral images were employed to validate the proposed approach. They can be seen in Figures 3–5. The three sets are symbolized as "T1," "T2," and "T3." Note that in each dataset, there were two scenes which had similar landscape patterns and geo-contents. For convenience, the two images in "T1" are coded as "T1A" and "T1B." "T1A" was used for AL execution, while "T1B" was exploited for validation experiments (the images in "T2" and "T3" were coded and used in the same way). To be more specific, AL was firstly applied to "T1A," leading to some of the samples that were selected from this image; these samples were then used for training an RF classifier that was subsequently employed for the classification of "T1B." The objective of such an experimental design was to see whether samples of good generalizability could be selected by using the proposed AL technique. If the samples selected from "T1A" could lead to high classification accuracy for "T1B," then we could conclude that these samples had sound generalizability, and, accordingly, the AL method had good performance.

The two images in "T1" were acquired by Gaofen-1 satellite, while the scenes in "T2" and "T3" were all acquired by Gaofen-2 satellite. The two Gaofen satellites are both Chinese remote sensing platforms. They were designed to capture earth-observing imagery with high quality. The sensors on-board Gaofen-1 and Gaofen-2 are similar, both including a multi-spectral and a panchromatic camera device. Their spectral resolutions are the same, while the two satellites differ in spatial resolution. Both of the multispectral sensors on-board Gaofen-1 and -2 have 4 spectral bands: near-infrared (NIR) (770–890 nm), red (630–690 nm), green (520–590 nm), and blue (450–520 nm). Their spatial resolutions (represented by ground sampling distance) are 8.0 and 3.24 m, respectively. In this study, the images were all acquired by the multi-spectral sensor.

"T1A" and "T1B" had the same acquisition date, which was 17 Nov 2018, but they were subsets that were extracted from different images. The sizes of "T1A" and "T1B" were 1090 × 818 and 1158 × 831 pixels, respectively. As can be seen in Figure 3, there were 2 large coal mine open-pits in "T1A," while there was only one in "T1B." The central-pixel coordinates for the two images were (E112°23′51″, N39°29′46″) and (E112°27′17″, N39°33′56″), respectively. This indicated that the geo-location of the two scenes was Antaibao, which is within Shuozhou city of Shanxi province, China. This place is the largest coal mine open-pit in China, and it has experienced extensive mining activities since 1984. It still keeps the highest daily production record, which is 79 thousand tons. However, the operational mining has produced significant ecological and hydrological effects on the local environment, so it is of environmental importance to monitor this area by using remote sensing techniques. Considering that the open-pit region has a large acreage, and it is inconvenient and even dangerous to collect field-visit data at this place. Therefore, it is meaningful to apply AL to the mapping task of this area.

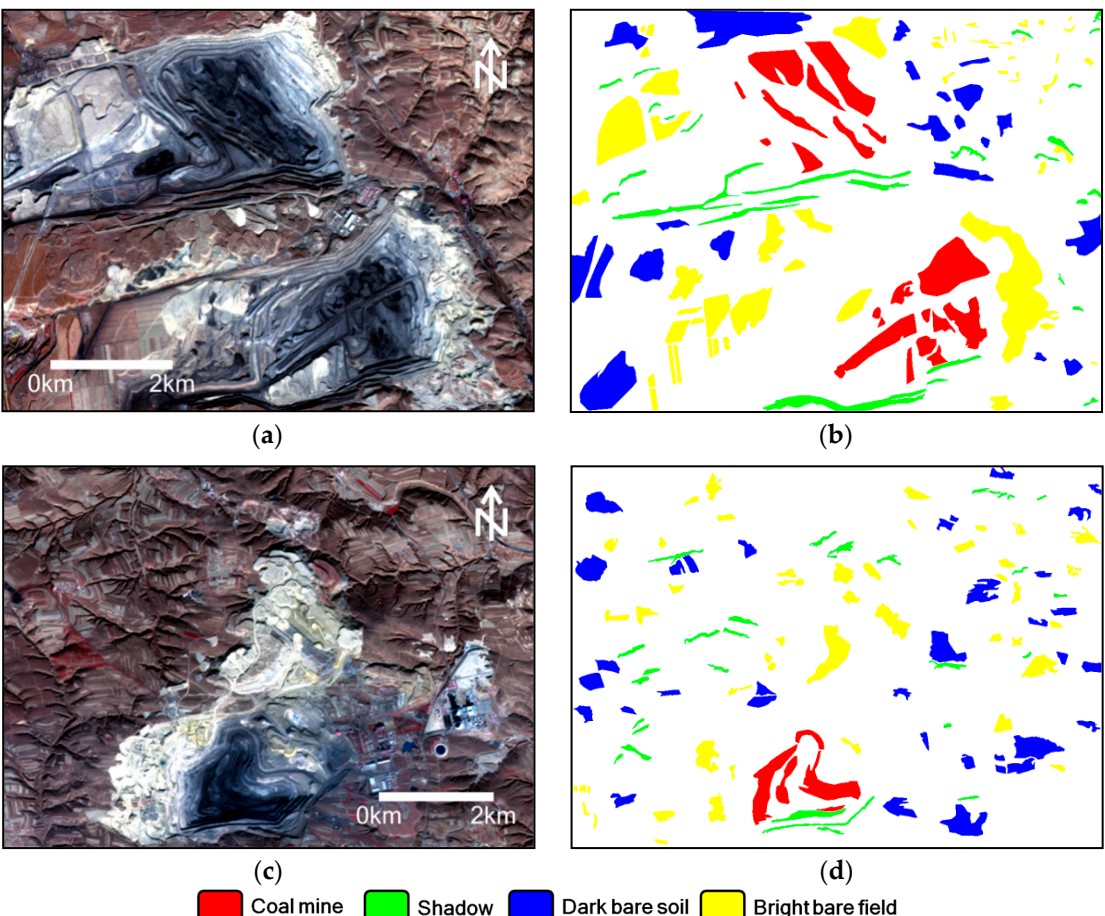

**Figure 3.** The dataset "T1." (**a**,**c**) are the original images of "T1A" and "T1B," respectively. Their color configuration is R: near-infrared (NIR); G: red; and B: green. (**b**,**d**) are the ground truth sample polygons. The legend at the bottom shows the class types of the samples.

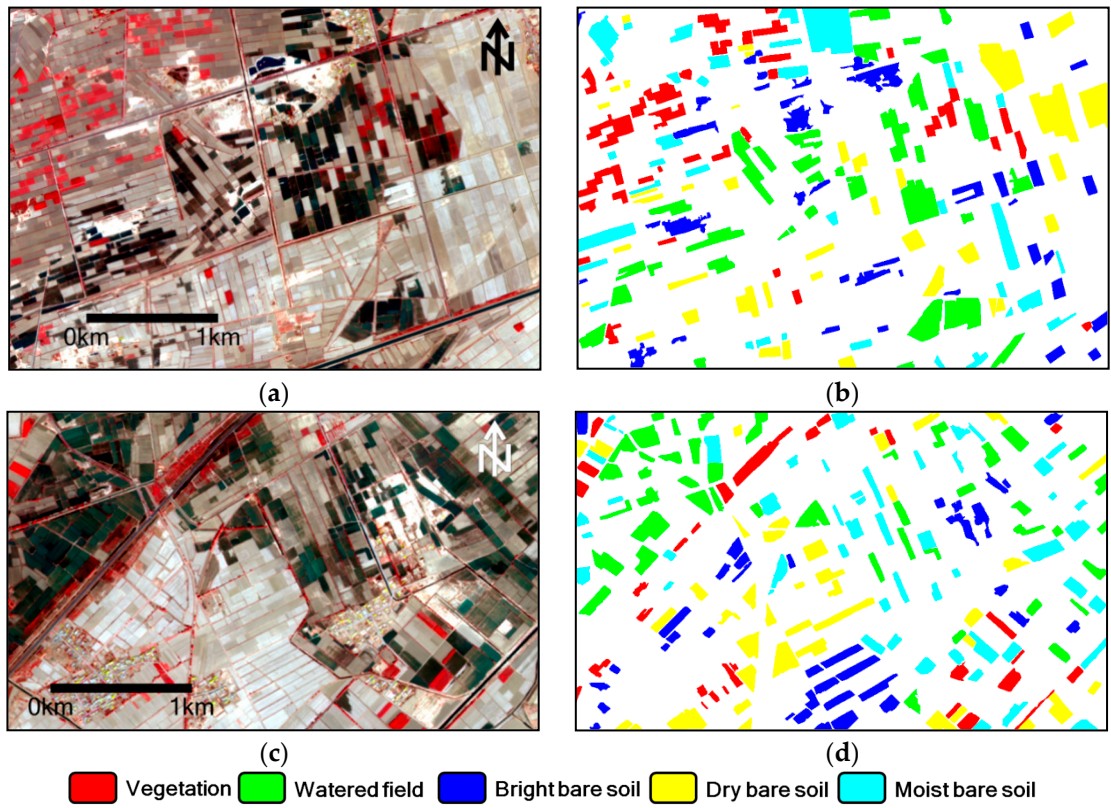

**Figure 4.** The dataset "T2." (**a**,**c**) are the original images of "T2A" and "T2B," respectively. Their color configuration is R: NIR; G: red; and B: green. (**b**,**d**) are the ground truth sample polygons. The legend at the bottom shows the class types of the samples.

The two images of "T2" were acquired on the same day, which was 7 May 2017. Similar to "T1," "T2A" and "T2B" were also subsets extracted from two different scenes. The sizes of the two subsets were, respectively, $1010 \times 683$ and $942 \times 564$ pixels. The central pixel coordinates of "T2A" and "T2B" were (E108°19′6″, N41°7′17″) and (E108°4′13″, N41°0′4″), respectively. According to this, both subsets were located at a county called Wuyuan Xian, which is in western Inner Mongolia, China. "T2" is illustrated in Figure 4, and it is evident that both subsets covered agricultural landscapes. Because the acquisition date was at the initial stage of the local agricultural calendar, many fields do not have vegetation cover. Instead, they are bare soil with different levels of moisture. Note that there are some immersed fields and damp ones. This is due to the practice of immersion, which reduces the saline and alkaline contents of local soil. In this way, farmers can grow crops such as corn, wheat, and sunflower at this place. Though local agriculture is quite developed, transportation in this rural region is still inconvenient, which increases the cost and difficulty of field data collection. This fact makes AL useful for mapping this place.

"T3" was very different from "T1" and "T2" in two aspects. First, the two images in "T3" covered urban areas. Second, the two subsets in "T1" or "T2" were quite close, while those of "T3" were very distant. The second aspect led to relatively large differences between the two "T3" subset images, which was conducive to testing the generalizability of the proposed AL. "T3A"/"T3B" had a size of 1069 $\times$ 674/995 $\times$ 649 pixels, and the central pixel coordinate was (E116°4′42″, N30°36′55″)/(E113°31′20″, N23°8′8″). The acquisition date of "T3A" was 2 Dec 2015, while it was 23 Jan 2015 for the other image. "T3A" captured an industrial area of Wuhan City, China, while "T3B" was in the economic development area of Huangpu District, Guangzhou City, China. Figure 5 exhibits the two images of "T3." Though at first glance, it is conspicuous that "T3A" and "T3B" had a similar urban appearance, their geo-objects were quite different in spatial distribution and quantity. There was more vegetation cover and more bright buildings in "T3B" than "T3A," and the vegetation in "T3B" was more reddish

than the counterpart of "T3A." Moreover, there were more light color buildings in "T3A" than "T3B." These differences were mainly due to the difference of geo-location and acquisition time.

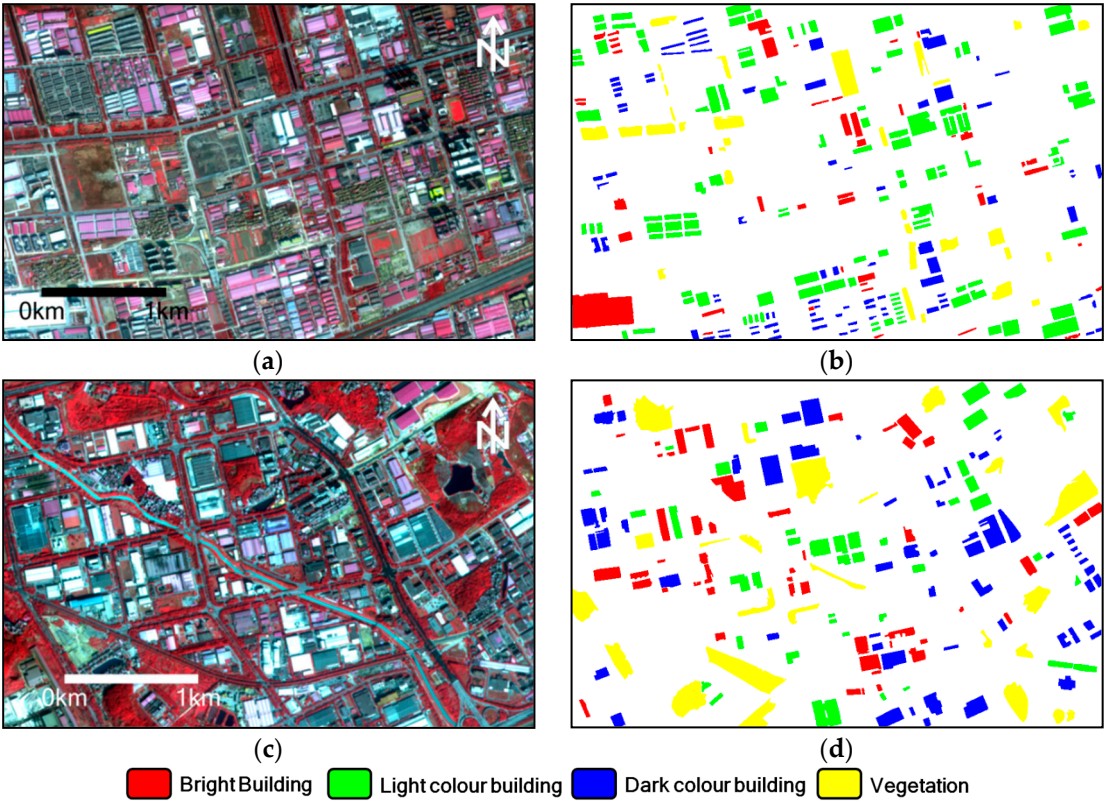

**Figure 5.** The dataset "T3." (**a**,**c**) are the original images of "T3A" and "T3B," respectively. Their color configuration is R: NIR; G: red; and B: green. (**b**,**d**) are the ground truth sample polygons. The legend at the bottom shows the class types of the samples.

For the three datasets, since the two subset images came from different scenes that differed in solar illumination and atmosphere conditions, the spectral signatures of the same land cover type may not have been consistent. To alleviate this effect, atmospheric correction was performed by using the quick atmospheric correction tool in ENVI 5.0 software.

The three datasets were made publicly available through using the following link so that readers can test other AL approaches by using the data of this work. The link is: https://pan.baidu.com/s/1I2jjLZvVqPZG4yEOjxdiyA.

## 3.2. Sample Collection

The AL-based classification experiment required training and testing samples. In this study, we determined these samples by using visual interpretation and manual digitization. For this purpose, we hired 3 experienced remote sensing image interpreters to extract geo-objects in the three datasets. In this process, they used polygons for digitization since this scheme can speed up sample collection. To guarantee the correctness of the collected samples, the interpreters cross-checked the initially obtained samples, and the polygons with high certainty and confidence were finally used in this experiment. The right columns of Figures 3–5 demonstrate the resulted reference samples. The numbers of the sampled polygons are listed in Tables 3–5. The following details the types of land use and land cover in the three image datasets.

**Table 3.** Reference samples collected for "T1.".

| Class Name | Coal Mine | Shadow | Dark Bare Soil | Bright Bare Soil |
|---|---|---|---|---|
| # polygons (pixels) for "T1A" | 20 (47,727) | 27 (18,082) | 21 (49,431) | 53 (67,455) |
| # polygons (pixels) for "T1B" | 5 (13,710) | 29 (10,658) | 35 (39,132) | 49 (31,962) |

**Table 4.** Reference samples collected for "T2.".

| Class Name | Vegetation | Watered Field | Bright Bare Soil | Dry Bare Soil | Moist Bare Soil |
|---|---|---|---|---|---|
| # polygons (pixels) for "T2A" | 34 (23,766) | 34 (42,043) | 39 (24,037) | 38 (33,769) | 36 (29,412) |
| # polygons (pixels) for "T2B" | 30 (13,317) | 44 (29,569) | 36 (18,509) | 39 (25,486) | 52 (24,835) |

**Table 5.** Reference samples collected for "T3.".

| Class Name | Bright Building | Light Color Building | Dark Color Building | Vegetation |
|---|---|---|---|---|
| # polygons (pixels) for "T3A" | 46 (17,587) | 111 (44,094) | 82 (15,601) | 32 (17,886) |
| # polygons (pixels) for "T3B" | 50 (18,665) | 40 (17,610) | 62 (27,841) | 25 (37,116) |

In "T1," there were four major geo-object types, including coal mine (red), shadow (green), dark bare soil (blue), and bright bare soil (yellow). The colors in brackets correspond to the sample map displayed in the right column of Figure 3. Coal mine and shadow were both dark and blackish in spectra, which was confusing and thus brought a great challenge to their differentiation. However, the geometric features of the 2 object types were quite different since most shadow areas were thin and elongated, while this pattern was not very evident for coal mine. The other 2 types were relatively easy to discriminate, mainly due to their distinctive spectral and textural appearance, but their spectral and textural features had a large range of variance, which may have confounded classification results.

As for "T2," we identified 5 land cover types, namely vegetation (red), watered field (green), bright bare soil (blue), dry bare soil (yellow), and moist bare soil (cyan). According to the local crop calendar, the vegetation in "T2" was mostly wheat since the other crop types such as corn and sunflower were not planted in early May. It is interesting to note that in the first half of May, the locals immerse the crop fields with irrigation water before sowing the seeds of sunflower or some vegetables, which leads to the dark-color fields found in "T2." The difference between watered field and moist field is that the former is covered by water, while the latter is merely soil with relatively high moisture. Different from dry bare soil, the bright bare soil fields are not used for growing crops. They are adopted for stacking harvested crops and are usually very flat, resulting in a high reflectance and, thus, a whitish appearance. The heterogeneous fields of vegetation, dry bare soil, and moist bare soil were not easy to discriminate, which can be recognized as a challenge for "T2"'s classification.

By carefully observing the two images of "T3," we determined 4 major land cover categories, which are bright building, light color building, dark color building, and vegetation. The 3 building types have different spectral appearances because their materials are not the same. The bright buildings mostly have flat cement roofs, while metal makes up the counterparts of the light color buildings. The dark color buildings correspond to brick-roofs, and their appearance is dark gray. Though there is a small lake and a thin river in "T3B," we did not consider this type since water objects do not exist in "T3A." Note that the vegetation in "T3A" consists mainly of bushes, meadows, or low trees, with small areas and light red color. In the other image, forests dominate the vegetation class, leading to a very reddish color. Such an inconsistency of spectra may have contributed to some classification errors for this dataset.

Note that in this experiment, for each dataset, we used the first image ("T1A," "T2A," or "T3A") to run AL, resulting in an enlarged set of training samples. The resulted sample set was then adopted to train an RF classifier, aiming to finish the classification task of the second image ("T1B," "T2B," or "T3B"). The ground truth samples of the second image were employed to evaluate classification performance,

while we exploited the samples of the first image for the labeling step of AL, which corresponds to the component *S* in Table 1. To be more specific, when running the AL method, some unlabeled samples were selected, which means that their labels were unknown to the AL algorithm. The ground truth samples were used for labeling these samples so that they could be added to the training set.

The aforementioned experimental process generally requires a large number of human–computer interactions, especially when many iterations in AL exist. To automate this procedure, we extracted the samples with labels in the first image and made their labels initially unknown to an AL approach (note that at the beginning of an AL, only a very small training set is inputted). When the AL determined some samples with good appropriateness, their labels were then inputted to the AL to enable the following AL steps. By doing so, we could automatically test an AL method with many repetitions, each of which was initialized by using a different initial training sample set. In this way, the effects of the initial sample configuration on AL performance could be investigated, and we hold that this plays a significant part in the validation of an AL algorithm.

Note that the AL method and the subsequent classification are all based on objects, i.e., the processing unit used in this study was a segment/object. However, the ground truth samples presented in Figures 3–5 are polygons containing non-overlapping pixels. To deal with this inconsistency, we needed to select the objects that match the ground truth samples. Considering that the object boundaries did not often align with those of sample polygons, we designed a matching criterion that is expressed by Equation (4),

$$H = \begin{cases} 1 \text{ if } \frac{n_{\mathrm{m}}}{n_{\mathrm{o}}} > T_{\mathrm{s}} \\ 0 \text{ otherwise} \end{cases} \tag{4}$$

where *H* is a test value for an object, and the object is selected only when it equals 1, $n_{\mathrm{m}}$ represents the maximum number of object pixels overlapped with the sample polygon(s) of one class, $n_{\mathrm{o}}$ means the number of pixels of the object under consideration, $T_{\mathrm{s}}$ is a user-defined threshold, and its numerical scope is (0,1). In this study, 0.7 was found to be sufficient for "T1A" and "T2A." While for "T3A," a small value, 0.3, was chosen because larger values led to too few selected objects. For a selected object, its label was identical to that of the $n_{\mathrm{m}}$ pixels.

This criterion guarantees that only the objects with relatively high homogeneity were adopted in AL execution. Because when an object is too heterogeneous, it tends to be inherently under-segmented, and this produces negative effects on the classifier's performance.

In the experiment, a small portion of the selected objects were used as the training set (*T*), and the rest of them were treated as the unlabeled set (*U*). When a query function (*Q*) computed the appropriateness measure for a sample in *U*, the label information of *U* was made unknown to the AL algorithm. Only when the AL selected some samples were the labels of them loaded into the AL, mimicking the human–computer interaction conducted by the supervisor *S*.

## 4. Experimental Results

### 4.1. Results of Image Segmentation

To produce segmentation results, we adopted the multi-resolution segmentation (MRS) [55] algorithm because many previous OBIA studies have reported that this method can produce satisfactory segmentation for high-resolution remote sensing imagery [19,58]. MRS is a bottom-up region merging technique, and it has three parameters: a shape coefficient, a compactness coefficient, and scale. The former two parameters affect the relative contribution of geometric and compactness heterogeneity in the merging criterion, and they were tuned and, respectively, set as 0.1 and 0.5 throughout this work. It is generally considered that scale is the most influential parameter for MRS because it controls the average size of the resulted objects. A large or small value of scale may lead to under- or over-segmentation error, which is not beneficial to classification performance. Therefore, how to optimally set scale is a key issue.

To objectively determine the optimal scale value for the six images in the three datasets, an unsupervised scheme proposed by Johnson and Xie [59] (JX) was used in this paper. The JX method employs weighted variance (WV) and Moran's index (MI) to reflect the relative quality for a series of segmentation results, which are produced by using a queue of scale values. In greater detail, WV and MI values are firstly normalized, and their sum, which is called the global score (GS), is used to measure the relative goodness for a scale value. The scale with the lowest GS is considered optimal. In this work, 20 values of scale, ranging from 10 to 200 with 10 being the incremental step, were used for JX analysis.

The analytical results of the JX method for the three datasets are presented in Figure 6. According to the lowest GS values marked by the black circles, 60 and 50 were chosen as the optimal scale for "T1A" and "T1B," respectively. For "T2A" and "T2B," the selected scales were both 80. As for "T3A" and "T3B," 50 and 60 were, respectively, determined. These values were used for the segmentation of the corresponding image. The segmentation results can be seen in Figures 7–9. The sub-figures (b) in the three figures demonstrate the objects that were selected by using the criterion of Equation (4). Visual observation can indicate that the selected objects agreed well with real geo-object boundaries. The quantities of the selected objects for different classes are listed in Tables 6–8. Note that since the processing unit is an object instead of a pixel in OBIA, the differences in the number of objects do not reflect spatial dominance for different classes, because objects may vary greatly in size so that if a class has a large number of the selected objects that have a relatively small average size, this class may not be spatially dominant.

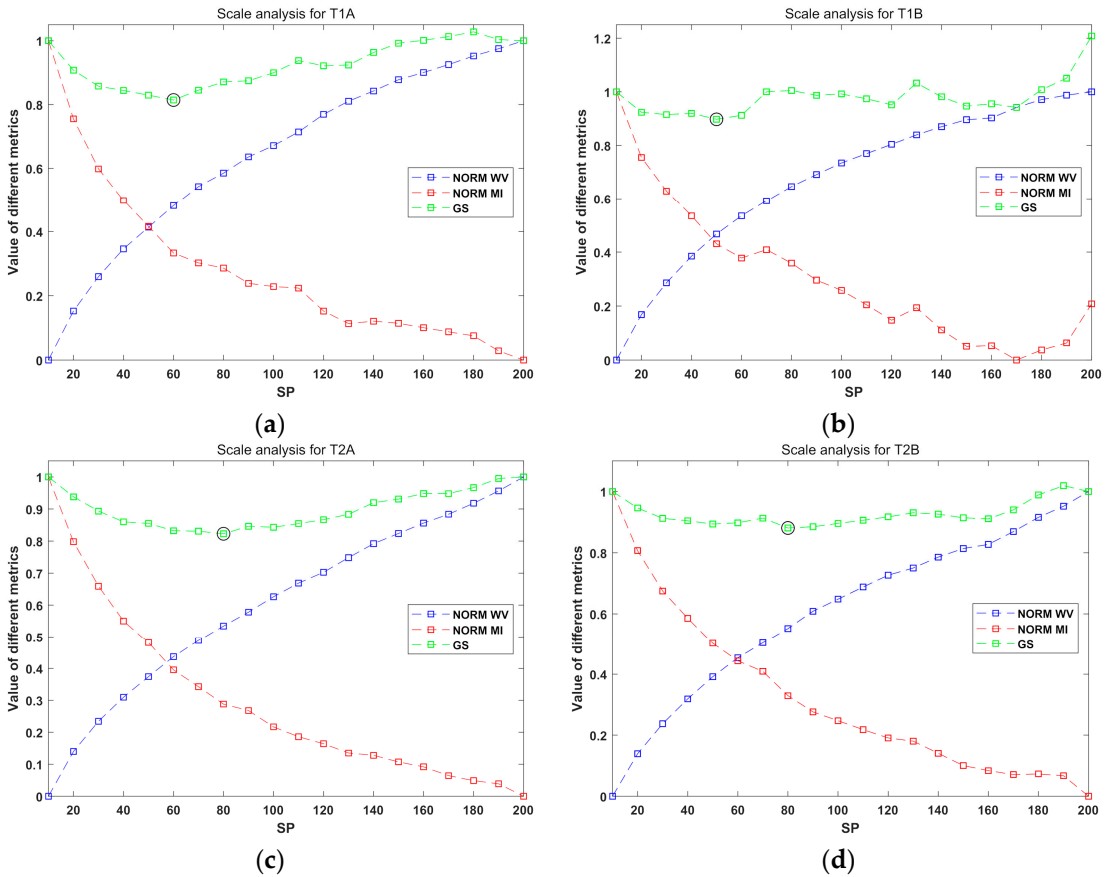

**Figure 6.** *Cont.*

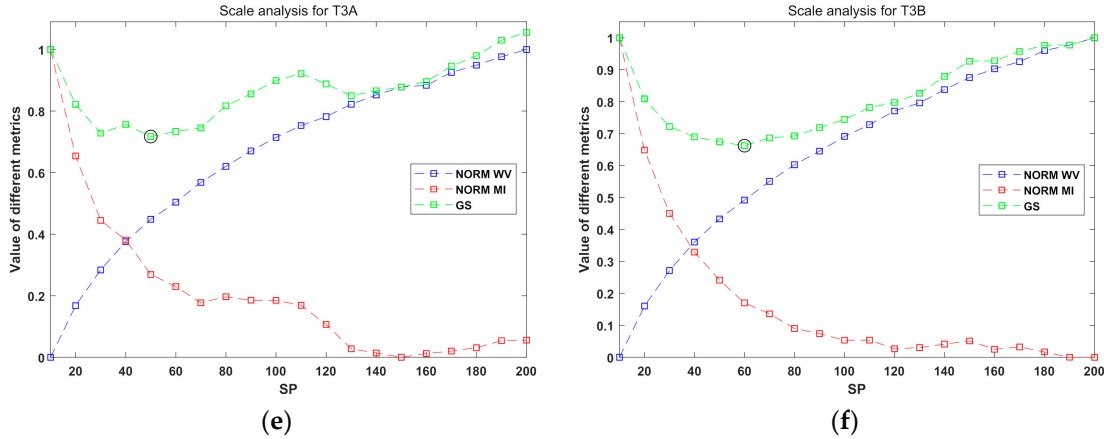

**Figure 6.** Scale analysis of the Johnson and Xie (JX) approach. (**a**) "T1A;" (**b**) "T1B;" (**c**) "T2A;" (**d**) "T2B;" (**e**) "T3A;" and (**f**) "T3B.".

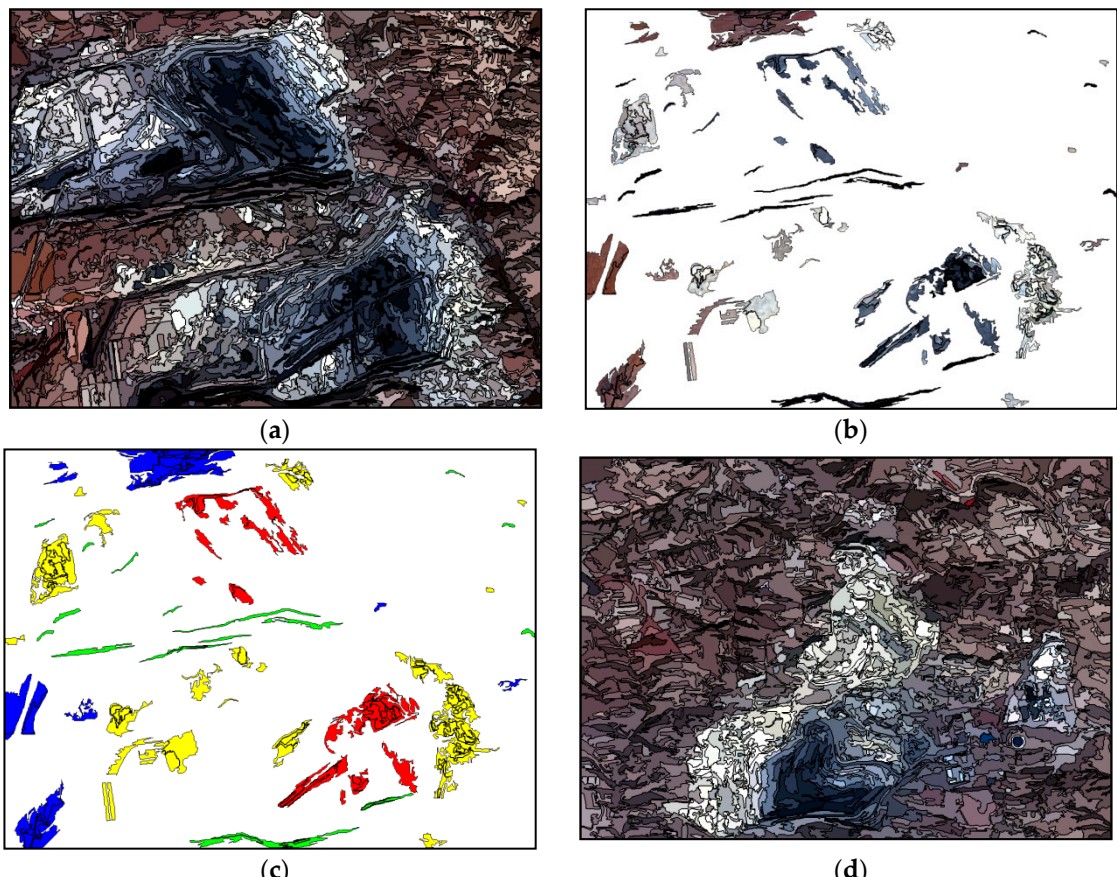

**Figure 7.** "T1"'s segmentation results and the selected samples used for the AL method. (**a**,**d**) are the segmentation results of "T1A" and "T1B," respectively. (**b**) shows the sample objects selected from "T1A," and (**c**) illustrates the class labels of the samples exhibited in (**b**).

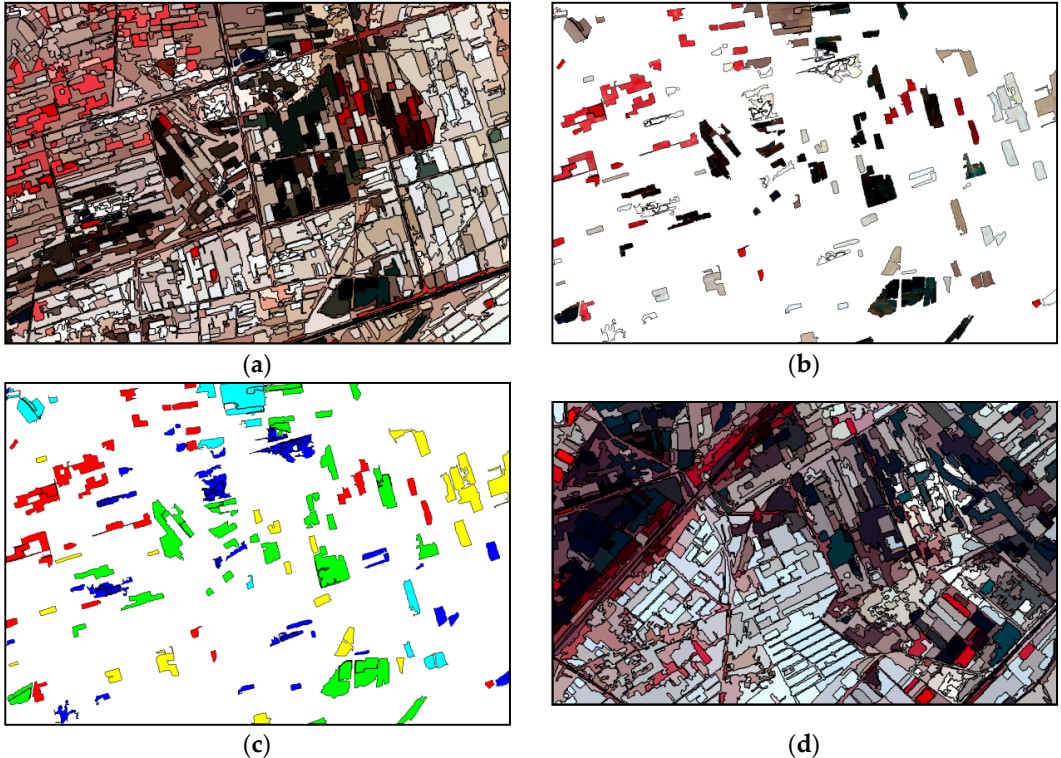

**Figure 8.** "T2"'s segmentation results and the selected samples used for AL method. (**a**,**d**) are the segmentation results of "T2A" and "T2B," respectively. (**b**) shows the sample objects selected from "T2A," and (**c**) illustrates the class labels of the samples exhibited in (**b**).

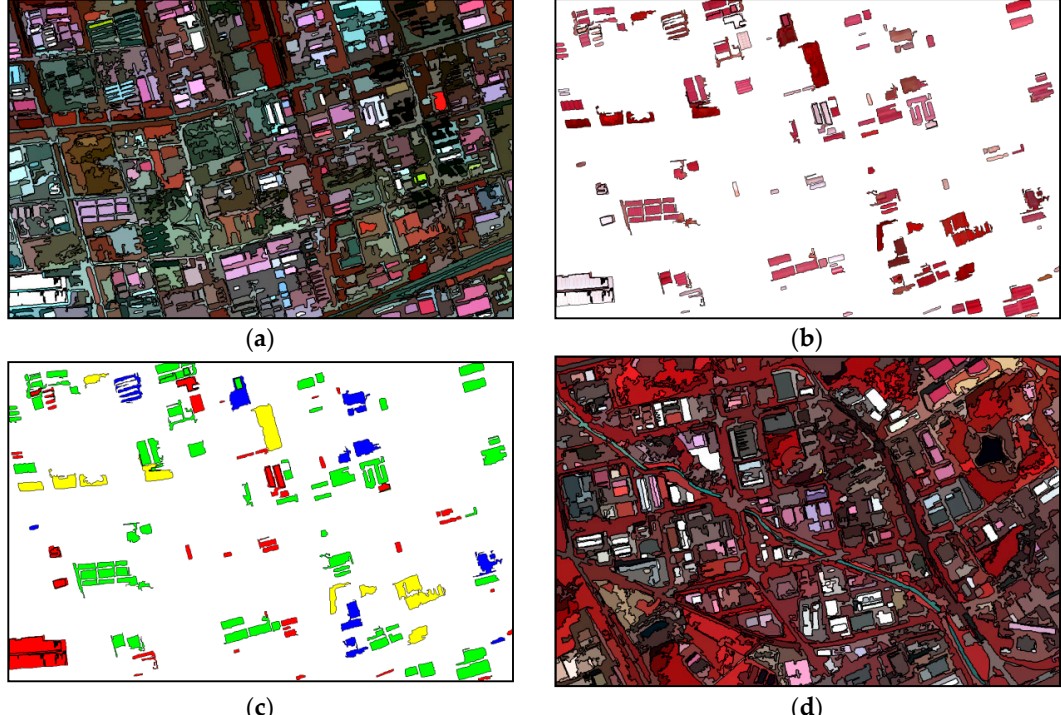

**Figure 9.** "T3"'s segmentation results and the selected samples used for AL method. (**a**,**d**) are the segmentation results of "T3A" and "T3B," respectively. (**b**) shows the sample objects selected from "T3A," and (**c**) illustrates the class labels of the samples exhibited in (**b**).

**Table 6.** Numbers of the selected objects for different classes in "T1A.".

| Class Name | Coal Mine | Shadow | Dark Bare Soil | Bright Bare Soil |
|---|---|---|---|---|
| # selected objects | 37 | 21 | 22 | 75 |

**Table 7.** Numbers of the selected objects for different classes in "T2A.".

| Class Name | Vegetation | Watered Field | Bright Bare Soil | Dry Bare Soil | Moist Bare Soil |
|---|---|---|---|---|---|
| # selected objects | 28 | 39 | 50 | 20 | 14 |

**Table 8.** Numbers of the selected objects for different classes in "T3A.".

| Class Name | Bright Building | Light Color Building | Dark Color Building | Vegetation |
|---|---|---|---|---|
| # selected objects | 51 | 56 | 10 | 10 |

### 4.2. Results of AL Experiment

With the selected objects mentioned in the last sub-section, AL can be initiated by choosing a portion of them as the training set $T$ and the left of them as the unlabeled set $U$. Different configurations of $T$ and $U$ can produce a great influence on classification accuracy, which affects the analysis of AL performance. To solve this issue, bootstrap analysis was used to generate several different $T$s and $U$s by using the entire selected objects. In doing so, 20 different bootstrap-acquired $T$s and 20 corresponding $U$s were repeatedly used for AL experiments. Each $T$ for "T1A" had 20 samples, and each class had five samples. For "T2A," the number of total samples in $T$ was 25, and the number of per-class samples was also five. There are 12 samples in each T of "T3A," with three samples for each class. The number of per-class samples for "T3A" was comparatively lower than those of the other two datasets because if a higher number was used, the increase of classification accuracy resulting from AL would have been very small, which was not conducive to analyzing the effects of AL.

To conduct classification by using the AL-selected training samples, RF was adopted, with the number of trees set as 300 and with the number of active features (also named as $m$ parameter in many previous studies [60,61]) set as the square root of the feature dimension. The aforementioned parameter configuration was used for the three datasets throughout the experiment, which was tested to be sufficient in this work.

To fully analyze the proposed method, five other AL strategies were implemented and used for comparison. The abbreviations of these approaches are provided in Table A2 of the Appendix A section. The details of M2, M3, and M4 are listed in Table 9. M1 is not included in Table 9 because it was introduced in Section 2. Note that in M2 and M3, the probability values of the unlabeled samples needed to be estimated, which was achieved by using the expectation-maximization method. To measure classification uncertainty, these two algorithms, respectively, used employ entropy and breaking tie criteria, which are frequently applied in AL research; thus, they can be recognized as state-of-the-art AL approaches. M4 randomly selects samples and acted as a baseline for the comparative study. M5 and M6 are both competitive methods and are constructed based on multinomial logistic regression. They were used in this experiment for comparison to further validate the proposed AL technique. M5 is a recently proposed method for hyperspectral imaging [62]. We modified this approach to enable it to process object-based samples. The novelty of M5 in terms of AL is its selective variance criterion. It has a parameter $V$, as described in [62]. In our experiment, this parameter was tuned and set as 0.1, which is consistent with the suggestion of the authors [62]. M6 is specifically designed for object-based AL, and its work-flow is very similar to M3 since M6 also uses a breaking tie criterion. The major difference between M2 and M6 is that the latter employs a multinomial logistic regression classifier to quantify probability values. Details of M6 can be found in [63,64]. We do not provide the working flow of M5 and M6 in Table 9 to save space. Interested readers can refer to the original articles for more details.

**Table 9.** Detailed processes of M2, M3, and M4.

**Input:** $T$, $C$, $U$, $S$, a threshold $T_{iter}$ ($T_{iter}$ works as a stopping criterion), and a parameter $q$ ($q$ determines how many samples are selected in each AL iteration)
**Output:** an enlarged $T$

| Process of M2 | Process of M3 | Process of M4 |
|---|---|---|
| 1. Let $N_{iter} = 1$; train an EM classifier ($C$) by using $T$; 2. Find $q$ sample(s) in $U$ by using the entropy query metric (Equation (17) of [39]); let $S$ provide label information for the $q$ sample(s); add the sample(s) into $T$; remove the sample(s) from $U$; $N_{iter} = N_{iter} + 1$; 3. Retrain $C$ by using the updated $T$; Go to step 2 if $N_{iter} < T_{iter}$; 4. Output $T$. | 1. Let $N_{iter} = 1$; train an EM classifier ($C$) by using $T$; 2. Find $q$ sample(s) in $U$ by using the entropy query metric (Equation (1) of [34]); let $S$ provide label information for the $q$ sample(s); add the sample(s) into $T$; remove the sample(s) from $U$; $N_{iter} = N_{iter} + 1$; 3. Retrain $C$ by using the updated $T$; Go to step 2 if $N_{iter} < T_{iter}$; 4. Output $T$. | 1. Let $N_{iter} = 1$; 2. Randomly select $q$ sample(s) from $U$; let $S$ provide label information for the $q$ sample(s); add the sample(s) into $T$; remove the sample(s) from $U$; $N_{iter} = N_{iter} + 1$; 3. Go to step 2 if $N_{iter} < T_{iter}$; 4. Output $T$. |

In the process of the six AL methods, $q$ unlabeled samples were iteratively selected and added to $T$. After each iterative step, the updated $T$ was used to classify "T1B," "T2B," or "T3B." An RF classification model was utilized for the classification task. The classification result was then assessed by using the reference samples shown in Figure 3d, Figure 4d, or Figure 5d. Overall accuracy (OA) calculated in a per-pixel fashion was adopted as the primary metric to reflect classification performance. Note that the OA that is obtained in the aforementioned way can only indicate the accuracy of the sampled area delimited by the polygons in Figure 3d, Figure 4d, or Figure 5d, not of the whole areas in "T1B," "T2B," or "T3B." This accuracy evaluation scheme was sufficient because the main objective of this work was to test AL performance instead of mapping the whole region of "T1B," "T2B," or "T3B."

The second objective of this work was to investigate the effects of different object-feature categories on the performance of object-based AL. For this purpose, eight different combinations of object feature sets were tested in the experiment. The eight cases are listed in Table A3 in the Appendix A. It can be seen that spectral features existed in all of the eight situations because this feature type is the most basic and most frequently used in OBIA. To the best of our knowledge, the fact that geometric features, textural features or contextual features are solely adopted for classification has never been encountered in previous studies, so the cases of G, T, and C are not presented in this work. For the convenience of analysis, the eight cases are divided into two groups, which are simple situations (including S, GS, ST, and SC) and complex situations (GST, GSC, STC, and GSTC). The former reveals whether geometric, textural, or contextual features can benefit the accuracy of object-based AL, while the latter indicates if the complex combination of different feature categories can produce a superior performance.

Note that for the six AL approaches, batch-mode was enabled. This means different numbers of samples (represented by $q$, as symbolized in Section 2.4) could be selected in each AL iteration. The effects of $q$ were thus investigated in the experiment to see how $q$ affects AL performance in OBIA.

4.2.1. Effects of Feature Combinations on AL Performance

The key aspect of AL is whether the selected training samples can increase classification accuracy. Figure 10 illustrates such effects of the four AL methods for "T1." As introduced above, 20 bootstrap-derived $T$s and $U$s extracted from "T1A" were used, leading to 20 different overall accuracies of "T1B" for each AL method in each iteration, and the average value was employed to plot Figure 10. $q$ was set as 5, as can be observed in the horizontal axis in Figure 10.

Among the eight situations of feature combinations, the proposed technique had better performance than the other five methods in the cases of S, GS, SC, GTS, GSC, and STC. Though the advantage of M1 was not very evident when GSTC was used, the proposed method led to a

higher average overall accuracy tan the other approaches when the number of training samples was between 40 and 55. Note that in the case of ST, M1's performance was not the best, and M3 had the most superior accuracy curve, followed by M6, whose accuracy curve was similar in shape to that of M3. It is interesting to note that for the four complex combinations, the accuracy curves of the six approaches had larger fluctuations than the counterparts of the four simple combinations. This implies that a complex feature space tends to bring larger challenges for an AL method. Another thing worth mentioning is that the best average overall accuracy was obtained by using M1 in the GS feature combination (88.32% when the number of training samples was 60), while the accuracies achieved in the complex combinations were not very advantageous. This further indicates that for object-based AL, it is not helpful to adopt a very complex feature space. The better accuracies of GS may have been due to the discriminative power of geometric features since coal mine and shadow objects had similar spectral appearances while differing in shape.

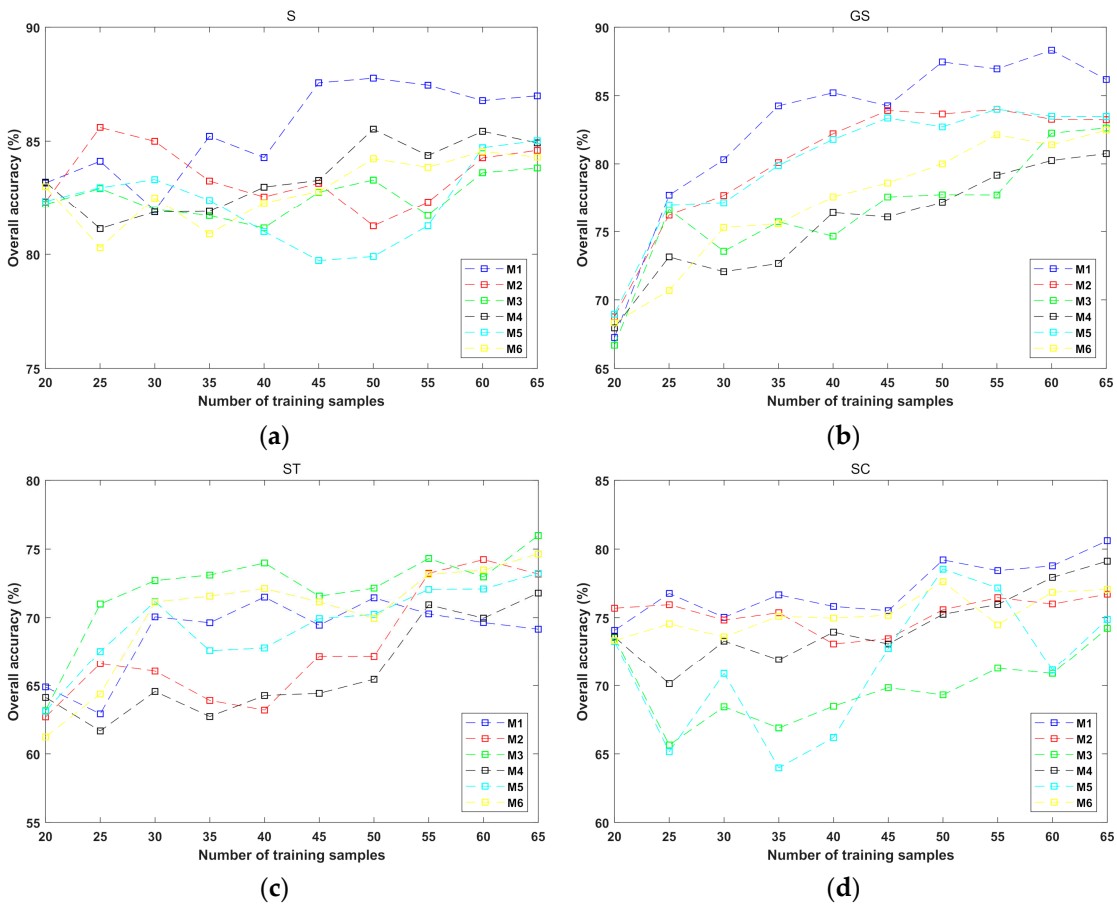

**Figure 10.** *Cont.*

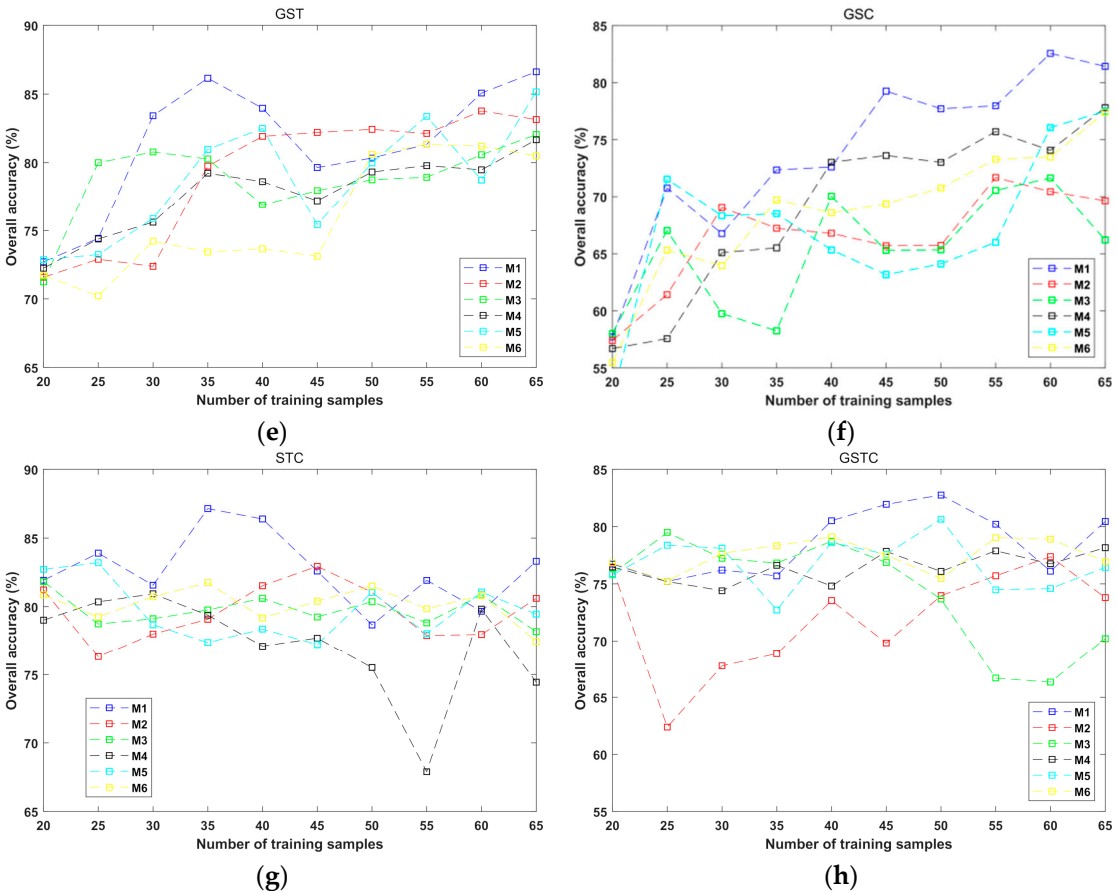

**Figure 10.** The dependence of AL performance of "T1" and the number of AL-selected training samples for the 4 AL algorithms in the 8 situations of object feature combinations. (**a**) S; (**b**) GS; (**c**) ST; (**d**) SC; (**e**) GST; (**f**) GSC; (**g**) STC; and (**h**) GSTC.

The analytical results of "T2" in the eight situations of feature combinations are illustrated in Figure 11. These figures were derived by using the similar experimental setup used for "T1," except that in the initial training set *T*, there were only three samples for each class. It could be seen that M1 outperformed the other approaches in the cases of S, ST, SC, GST, GSC, and STC. Unlike the results of "T1," where M1 had the most conspicuous advantage for the GS feature combination, the superiority of M1 was not very evident in the case of GS for "T2," although the proposed technique did have better average overall accuracies when the number of training samples was in the ranges of [20,25,35,50]. As for the case of GSTC, M4 had the poorest performance, while M1, M2, M3, and M6 had similar accuracy curves. Note that in this feature combination, the average overall accuracy of M1 was the highest when the number of training samples was 55 and 60.

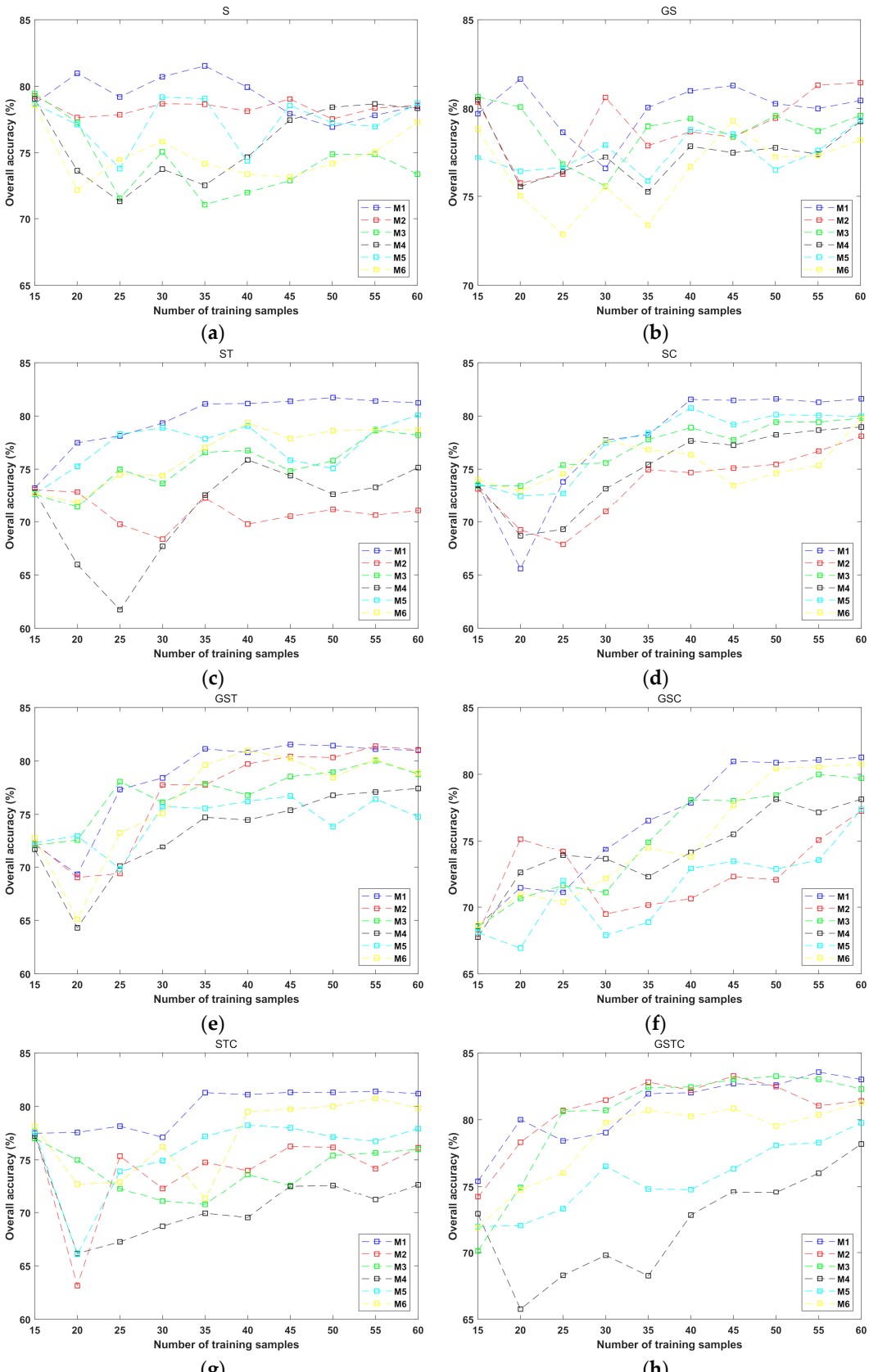

**Figure 11.** The dependence of AL performance of "T2" and the number of AL-selected training samples for the 4 AL algorithms in the 8 situations of object feature combinations. (**a**) S; (**b**) GS; (**c**) ST; (**d**) SC; (**e**) GST; (**f**) GSC; (**g**) STC; and (**h**) GSTC.

The performance of the six AL methods in the eight-feature-combination situations for "T3" is plotted against the number of the selected training samples, as exhibited in Figure 12. The experimental setup was similar to those for "T1" and "T2," while in the initial training set of "T3A," the number of per-class samples was two. It can be seen from Figure 12 that large variations and fluctuations existed for different AL methods and different feature combinations. Such an unstable performance was mostly due to the inconsistency of "T3A" and "T3B" since their geo-locations and acquisition times were very different. Among the eight feature combinations, M1 could be deemed to perform better in the cases of S, GS, SC, and GSC. For GSTC, although the accuracy curves of M1, M4, and M6 were quite close, the highest average accuracy value corresponded to the proposed technique when the number of training samples equaled 53. Note that for the eight cases and the six AL schemes, the average overall accuracies slumped when the number of the selected samples was within [23,28]. This was an interesting phenomenon, and its explanation may be that when the number of training samples increased from 23 to 28, all of the approaches selected some highly confusing samples from "T3A," which contributed to the performance degradation. Though the STC feature set resulted in poor performance, M1 still produced the best overall accuracy in this case, and it occurred in the first iteration of AL. In other words, the optimal accuracy of "T3B" (93.12%) was achieved by using M1 and STC when the number of training samples was 13. For this reason, the optimal classification results derived by using the six AL approaches are further analyzed in the next sub-section.

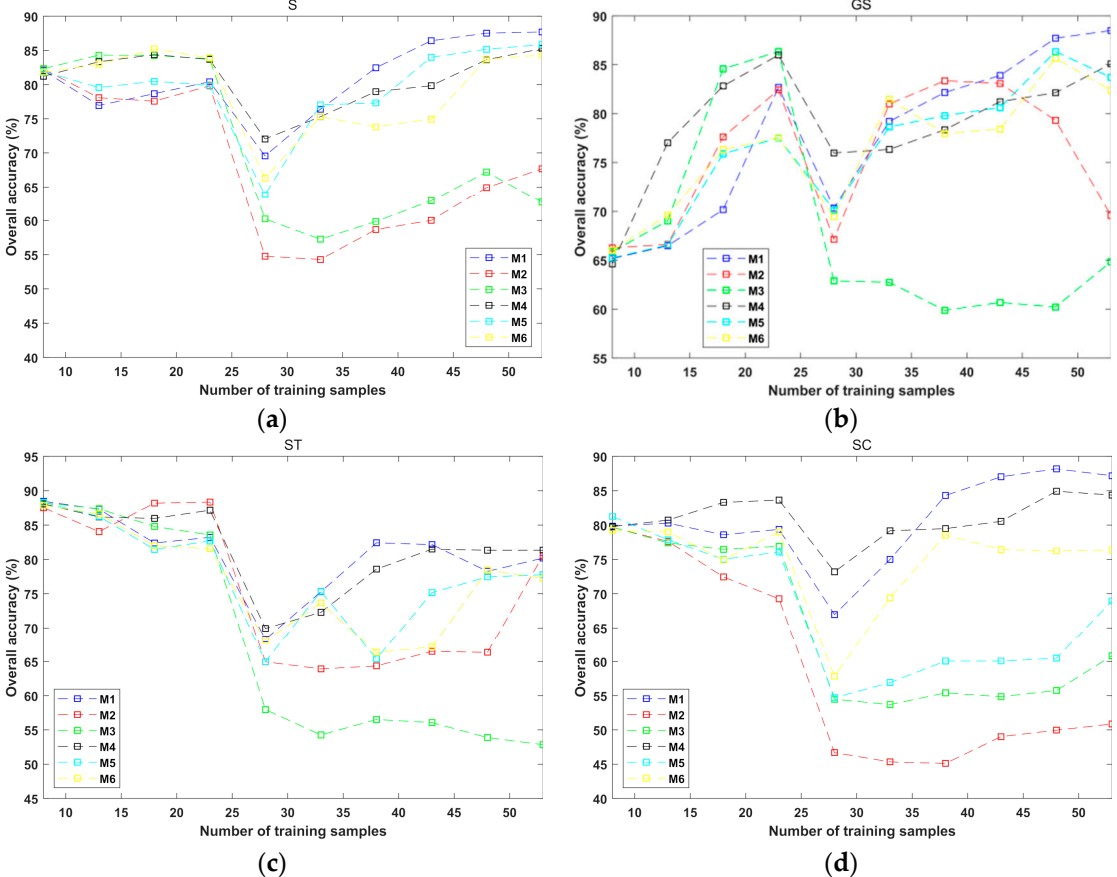

**Figure 12.** *Cont.*

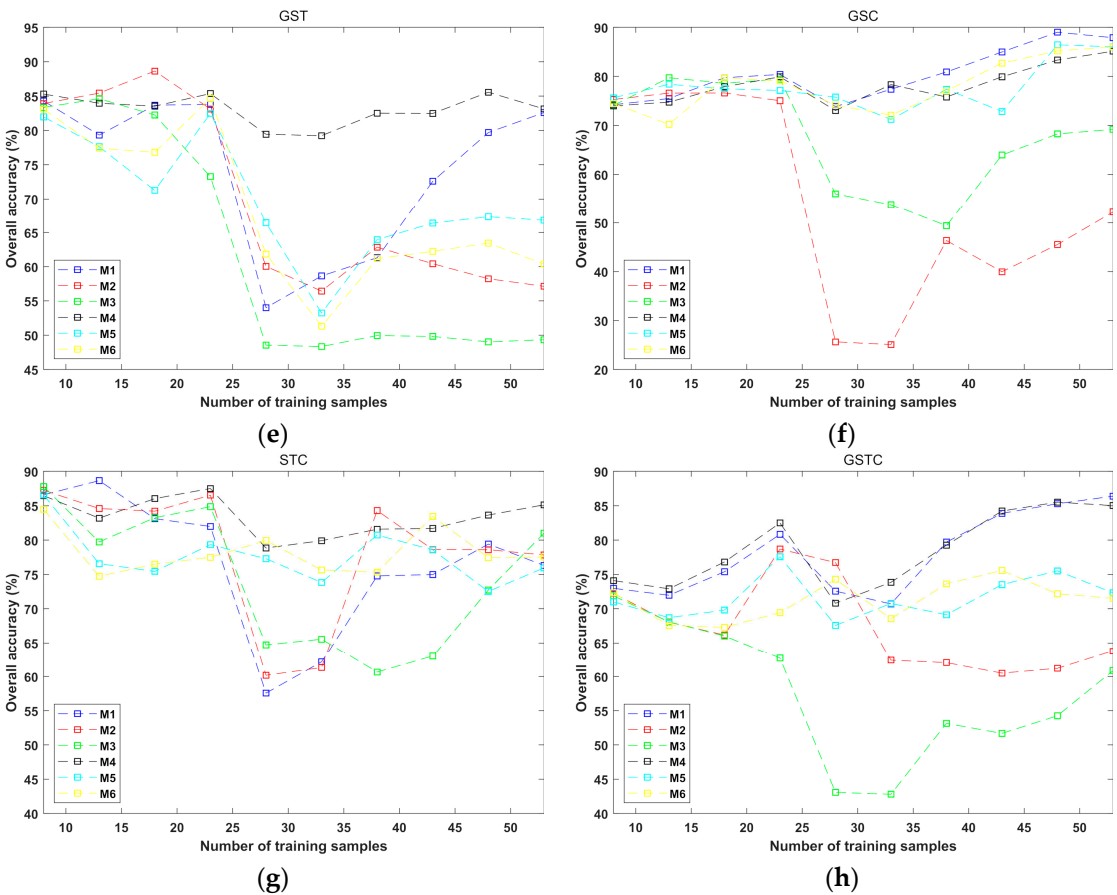

**Figure 12.** The dependence of AL performance of "T3" and the number of AL-selected training samples for the 4 AL algorithms in the 8 situations of object feature combinations. (**a**) S; (**b**) GS; (**c**) ST; (**d**) SC; (**e**) GST; (**f**) GSC; (**g**) STC; and (**h**) GSTC.

## 4.2.2. Comparison of the AL Classifications

To further compare the six AL methods for "T1," the best classification results that were obtained by using the six AL algorithms in the case of GS are displayed in Figure 13. The accuracy values, including class-specific accuracy metrics (user accuracy (UA) and producer accuracy (PA)) and overall accuracy, are provided in Table 10. For a fair comparison, the six results shown in Figure 13 were obtained by using the same initial training sample set *T*. For illustrative purposes, the classification errors are marked by using circles, and the color of a circle indicates the correct class type of the corresponding object. From Figure 13, it can be seen that there were fewer errors for coal mine in the results of M1, but in Figure 13d–f, many dark bare soil objects were wrongly assigned to other classes. Note that some bright bare soils were mistakenly classified by M1, as pointed out by the yellow circles. Similar errors could be observed in the results of M2, M3, M4, and M6. According to the classification maps in Figure 13, the spatial distributions of the classification errors were very different for the six AL methods, although the overall accuracies of M3 and M4 were quite similar. This was further supported by the UAs and PAs listed in Table 10 because these values varied greatly for the six methods. M1 had the best overall accuracy of 90.88%. This indicates that the samples that were selected by using the proposed AL algorithm had the best generalization ability.

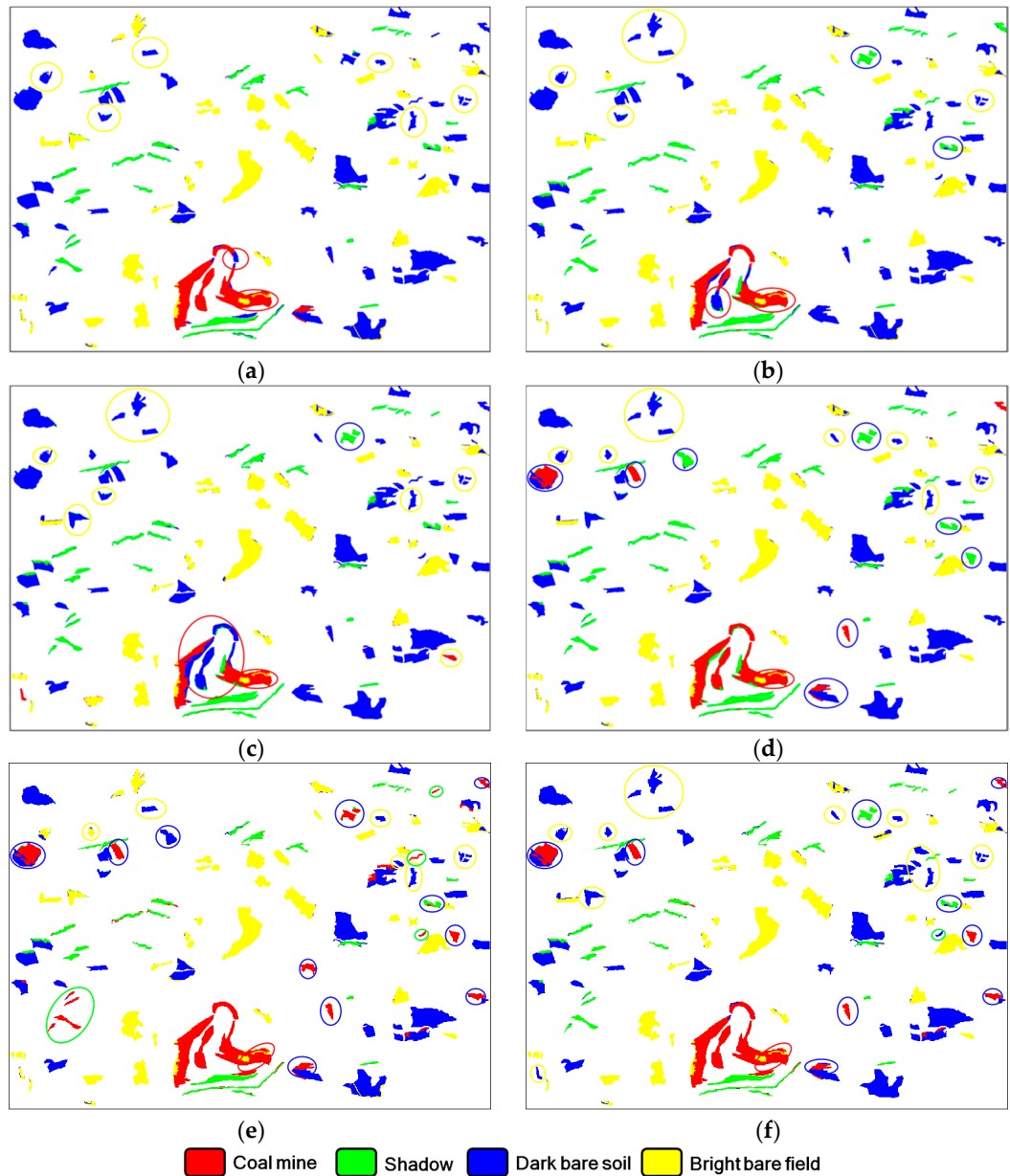

**Figure 13.** The optimal classification results of the 4 AL methods for "T1" when the GS feature combination was used. (**a**) M1; (**b**) M2; (**c**) M3; (**d**) M4; (**e**) M5; and (**f**) M6. The circles indicate the erroneously classified objects, and the color of a circle represents the correct class type for the corresponding object.

**Table 10.** Class-specific and overall accuracies of the 6 methods for "T1." The values of this table correspond to the classification results in Figure 13.

| Method | M1 | M2 | M3 | M4 | M5 | M6 |
|---|---|---|---|---|---|---|
| UA of coal mine (%) | 94.439 | 99.06 | 91.06 | 73.07 | 58.11 | 70.64 |
| UA of shadow (%) | 90.92 | 72.10 | 74.60 | 65.56 | 88.96 | 82.31 |
| UA of dark bare soil (%) | 87.45 | 83.52 | 76.27 | 83.43 | 91.20 | 81.03 |
| UA of bright bare soil (%) | 94.21 | 94.17 | 95.79 | 95.78 | 94.31 | 95.53 |
| PA of coal mine (%) | 84.8505 | 69.53 | 49.18 | 79.50 | 91.98 | 91.00 |
| PA of shadow (%) | 89.68 | 95.05 | 93.27 | 97.46 | 80.70 | 93.76 |
| PA of dark bare soil (%) | 94.53 | 91.53 | 94.49 | 80.67 | 77.42 | 81.94 |
| PA of bright bare soil (%) | 89.41 | 85.15 | 78.68 | 80.50 | 91.11 | 77.96 |
| Overall accuracy (%) | 90.88 | 86.63 | 82.56 | 82.32 | 84.46 | 83.23 |

To further analyze the results of "T2," we provide the optimal classification maps that were obtained by using the six AL strategies in Figure 14. The feature combination was GST, since in this case, M1 produced the highest overall accuracy for "T2" at 85.77%. Note that the six classification results were derived by using the same initial training set so that the comparison could be considered fair. At first glance, it could be seen that there were fewer errors in the results of M1, M2, and M6 than those of the other three methods. This agreed well with the accuracy values provided in Table 11 since the overall accuracies of M1, M2, and M6 were higher than those of others, which were all below 80%. The inferior performance of M3, M4, and M5 was partly due to some vegetation mistakenly being assigned to the watered field. A careful examination of the six maps indicated that the confusion between bright bare soil and dry bare soil was common for all of the six approaches. The low UAs for dry bare soil and the low PAs for bright bare soil for the six methods, as shown in Table 11, agreed well with this type of error. This phenomenon can be attributed to the similar spectral appearance of the two classes.

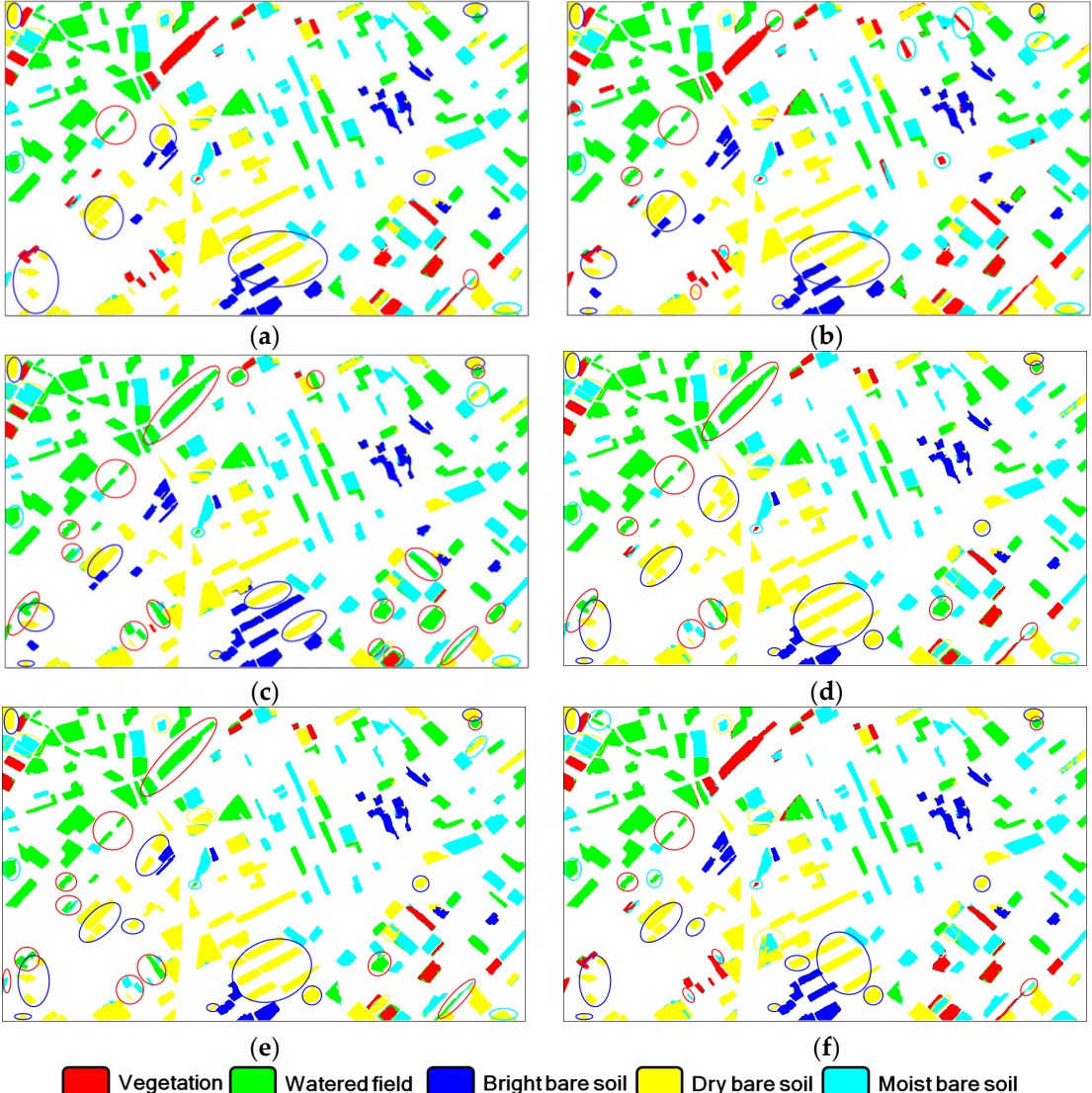

**Figure 14.** The optimal classification results of the 4 AL methods for "T2" when the GST feature combination was used. (**a**) M1; (**b**) M2; (**c**) M3; (**d**) M4; (**e**) M5; and (**f**) M6. The circles indicate the erroneously classified objects, and the color of a circle means the correct class type for the corresponding object.

**Table 11.** Class-specific and overall accuracies of the 6 methods for "T2." The values of this table correspond to the classification results in Figure 14.

| Method | M1 | M2 | M3 | M4 | M5 | M6 |
|---|---|---|---|---|---|---|
| UA of vegetation (%) | 96.71 | 81.44 | 100.0 | 98.83 | 100.00 | 95.51 |
| UA of watered field (%) | 91.65 | 90.10 | 71.34 | 79.34 | 78.15 | 91.18 |
| UA of bright bare soil (%) | 100.0 | 100.0 | 99.08 | 100.0 | 100.00 | 100.00 |
| UA of dry bare soil (%) | 69.37 | 71.96 | 75.67 | 62.42 | 64.14 | 68.83 |
| UA of moist bare soil (%) | 88.10 | 91.26 | 86.07 | 79.22 | 84.64 | 80.30 |
| PA of vegetation (%) | 81.33 | 83.65 | 17.98 | 42.42 | 38.92 | 82.17 |
| PA of watered field (%) | 98.04 | 97.42 | 98.64 | 97.28 | 98.06 | 97.24 |
| PA of bright bare soil (%) | 49.95 | 58.71 | 66.89 | 32.30 | 36.09 | 48.30 |
| PA of dry bare soil (%) | 94.20 | 94.20 | 90.41 | 85.80 | 92.35 | 83.40 |
| PA of moist bare soil (%) | 88.82 | 80.25 | 88.34 | 91.66 | 88.82 | 93.45 |
| Overall accuracy (%) | 85.15 | 84.81 | 79.60 | 76.11 | 77.39 | 83.33 |

In Figure 15, the classification maps of "T3" mentioned in the last sub-section are displayed. To ensure a fair comparison, the same initial training set *T* was used to produce these classification results. Table 12 provides the class-specific and overall accuracy values corresponding to the six maps. It can be seen that the maps in Figure 15a,d had comparable performances, and the spatial distributions of their classification errors were mostly similar. In the two result maps, it could be noticed that the area of the erroneously classified objects was small, while in the counterparts of the other four methods, more large-area objects were wrongly classified, such as the bright building objects in the middle and middle-bottom part of "T3B." This may have been the primary reason for the inferior accuracies obtained by using M2, M3, M5 and M6. The low PA values for light color building that were obtained by using M2, M3, M5, and M6, as can be seen from Table 12, further support this statement.

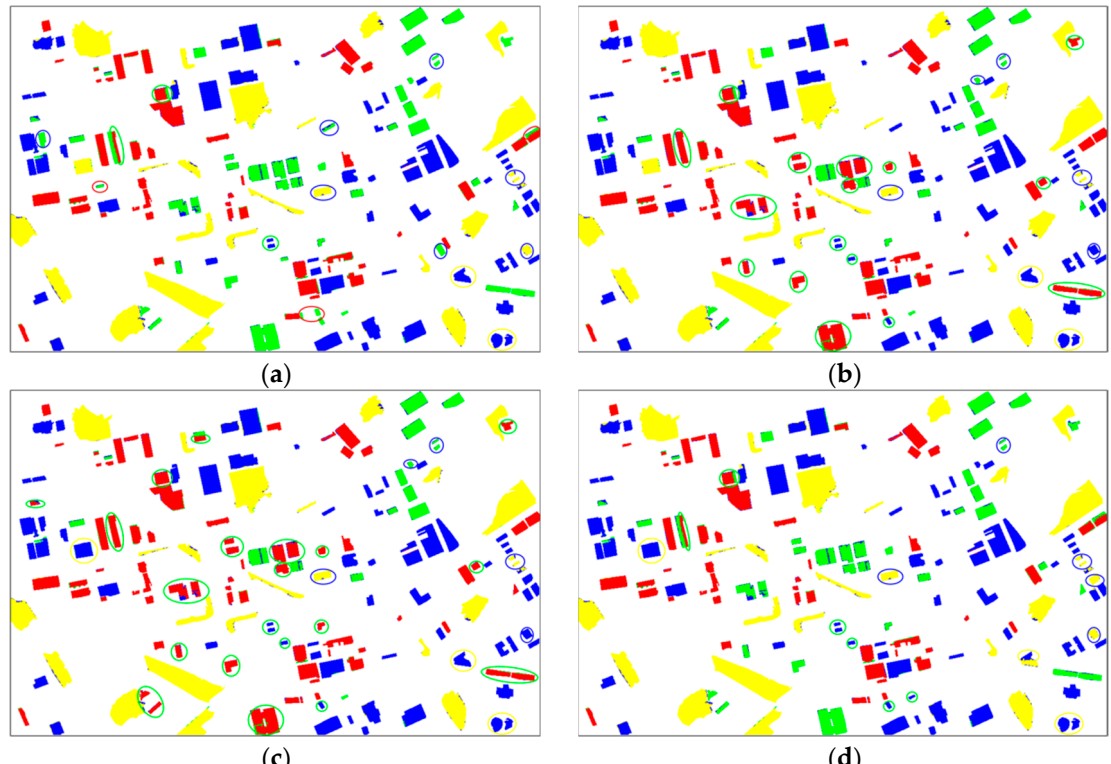

(**a**)　　　　　　　　　　　　　　　　　(**b**)

(**c**)　　　　　　　　　　　　　　　　　(**d**)

**Figure 15.** *Cont*.

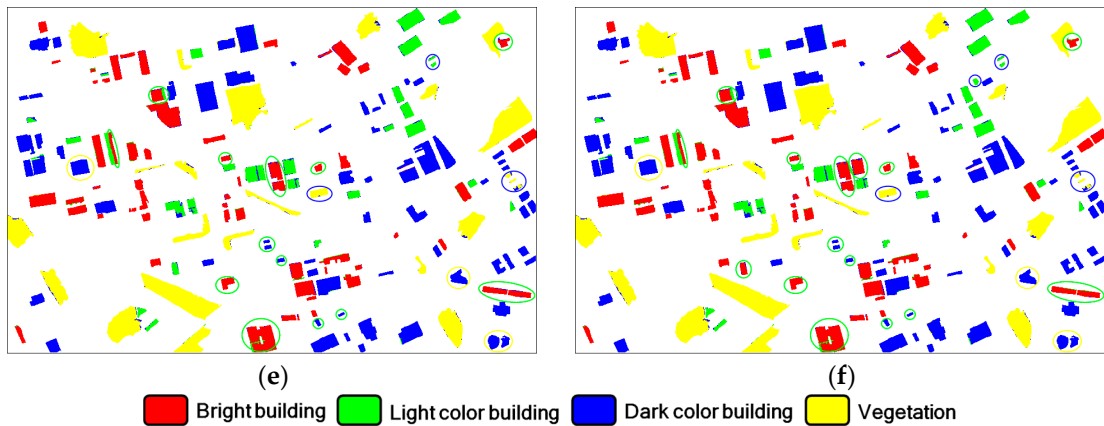

**Figure 15.** The optimal classification results of the 4 AL methods for "T3" when the STC feature combination was used. (**a**) M1; (**b**) M2; (**c**) M3; (**d**) M4; (**e**) M5; and (**f**) M6. The circles indicate the erroneously classified objects, and the color of a circle means the correct class type for the corresponding object.

**Table 12.** Class-specific and overall accuracies of the 6 methods for "T3." The values of this table correspond to the classification results in Figure 15.

| Method | M1 | M2 | M3 | M4 | M5 | M6 |
|---|---|---|---|---|---|---|
| UA of bright building (%) | 95.67 | 68.50 | 65.87 | 95.73 | 76.09 | 73.60 |
| UA of light color building (%) | 88.56 | 92.74 | 91.12 | 94.59 | 94.32 | 93.09 |
| UA of dark color building (%) | 89.93 | 88.76 | 86.84 | 86.98 | 86.16 | 86.32 |
| UA of vegetation (%) | 96.55 | 97.32 | 97.51 | 95.61 | 97.72 | 97.61 |
| PA of bright building (%) | 93.38 | 96.68 | 96.88 | 94.82 | 96.24 | 96.24 |
| PA of light color building (%) | 89.74 | 45.21 | 39.64 | 87.05 | 59.74 | 55.61 |
| PA of dark color building (%) | 91.71 | 95.11 | 95.41 | 93.42 | 96.02 | 95.57 |
| PA of vegetation (%) | 95.66 | 95.63 | 93.35 | 94.38 | 93.31 | 93.31 |
| Overall accuracy (%) | 93.12 | 86.91 | 85.22 | 92.92 | 88.76 | 87.91 |

### 4.2.3. Analysis of Parameter q

To investigate how $q$ affected the proposed AL approach, a series of $q$ values ($q = 1,3,5,7,9$) were tested for "T1." For each $q$, 20 different bootstrap-derived $T$s and $U$s were employed for repetitive runs. To eliminate the influence of different feature combinations, GSTC was adopted. The average overall accuracies are plotted in Figure 16. From this graph, it can be observed that, although some fluctuations existed, all of the accuracy curves had a similar trend, which was that more training samples led to higher accuracies. However, there were some inconsistencies for different $q$s when the number of training samples was larger than 45. Such a phenomenon may be ascribed to the differences of "T1A" and "T1B" since they were obtained from different Gaofen-1 scenes.

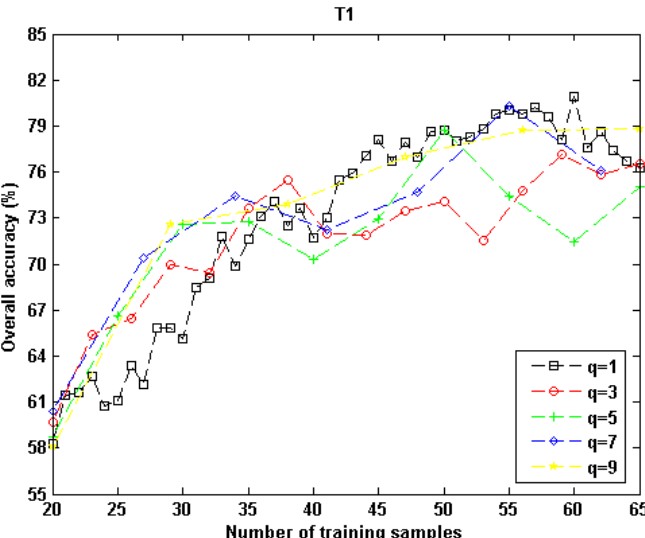

**Figure 16.** Effects of parameter *q* for the proposed AL algorithm in "T1"'s experiment. The feature combination was GSTC.

The effects of *q* on the proposed AL scheme were also analyzed by using "T2," and the results are shown in Figure 17. Similar to what is displayed in Figure 16, the accuracy curves of different *q* values had a similar trend. Contrary to the results of "T1," in which the average accuracy values of different *q*s had large discrepancies when the number of training samples was high, Figure 17 shows that the accuracy values became very close when the number of training samples was greater than 35. This indicates that *q* had a little influence on M1's performance for "T2" when a certain number of samples were selected. It is interesting to note that when the number of training samples was between 25 and 35, the highest accuracy values were achieved, and this pattern was the most conspicuous when *q* = 5,7,9. However, as the number of training samples increased, all of the accuracy curves became steady, which implies that the proposed technique was relatively robust in the batch AL mode for "T2."

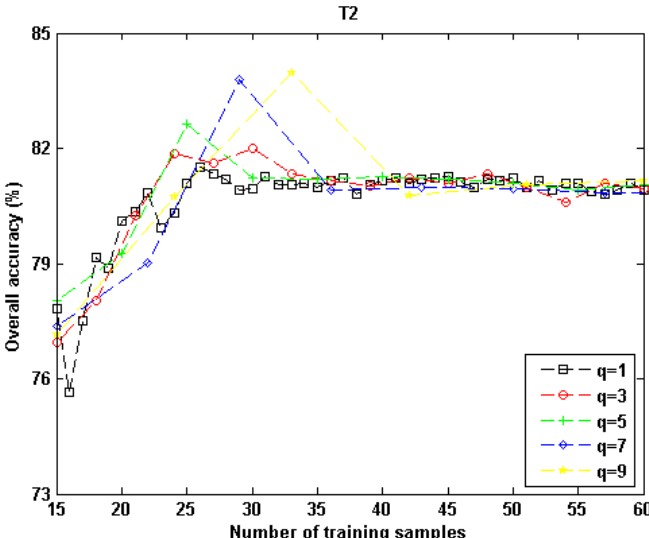

**Figure 17.** Effects of parameter *q* for the proposed AL algorithm in "T2"'s experiment. The feature combination was GSTC.

The effects of parameter *q* on the performance of the proposed algorithm were also analyzed for "T3," and the results are illustrated in Figure 18. Similar to what was found for "T1" and "T2," the accuracy curves of different *q*s increased as the number of samples rose. The peak values of these

curves all occurred when the number of samples was high (≥43). The differences in the accuracy curves became smaller with the increase of samples. These experimental results indicated that for "T3," the influence of $q$ on the performance of M1 was small when the number of samples was greater than 38.

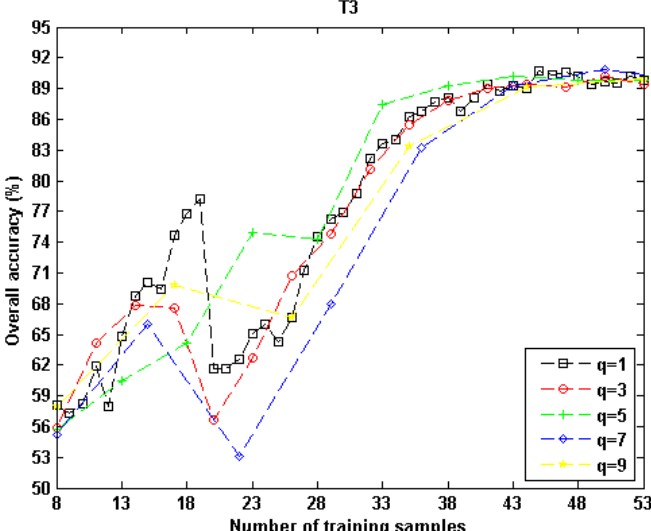

**Figure 18.** Effects of parameter $q$ for the proposed AL algorithm in "T3"'s experiment. The feature combination was GSTC.

## 5. Discussion

There were two objectives in this work. First, a new object-based active learning (AL) algorithm based on a binary random forest model was developed. To deal with the multiclass classification problem, the one-against-one strategy was adopted to better measure the classification uncertainty for the unlabeled samples in various situations. This aimed at more accurately estimating classification uncertainty so that more effective samples could be chosen during the AL process. Second, four different categories of object-based features were tested in the experiment to investigate whether AL performance could be affected by the change of feature combination. According to previous literature on AL in remote sensing, this aspect has been rarely analyzed, but we think that it is a meaningful research line, because feature space in OBIA is generally complex and may have a large influence on classification accuracy and consequently, on AL performance.

According to the objectives stated above, it was necessary to discuss the experimental results with a deeper analysis based on the information presented in Figures 10–12. This was achieved by comparing the learning rates of different AL methods. Intuitively, the learning rate was considered as the improvement of classification accuracy obtained by using an AL approach. In this work, the learning rate was calculated as the difference between the AL-free average overall accuracy and the highest average overall accuracy derived by using an AL method. Figure 19 provides the learning rates of the six AL algorithms with eight different feature combinations. It was straightforward to see that for the three datasets, M1 had the best learning rates in all feature-combination cases, except for the GSTC case of "T2," but it could be seen that the learning rates of the six AL methods were quite similar in this case. These results prove that the proposed AL technique can effectively improve the classification accuracy for the three high-resolution datasets used.

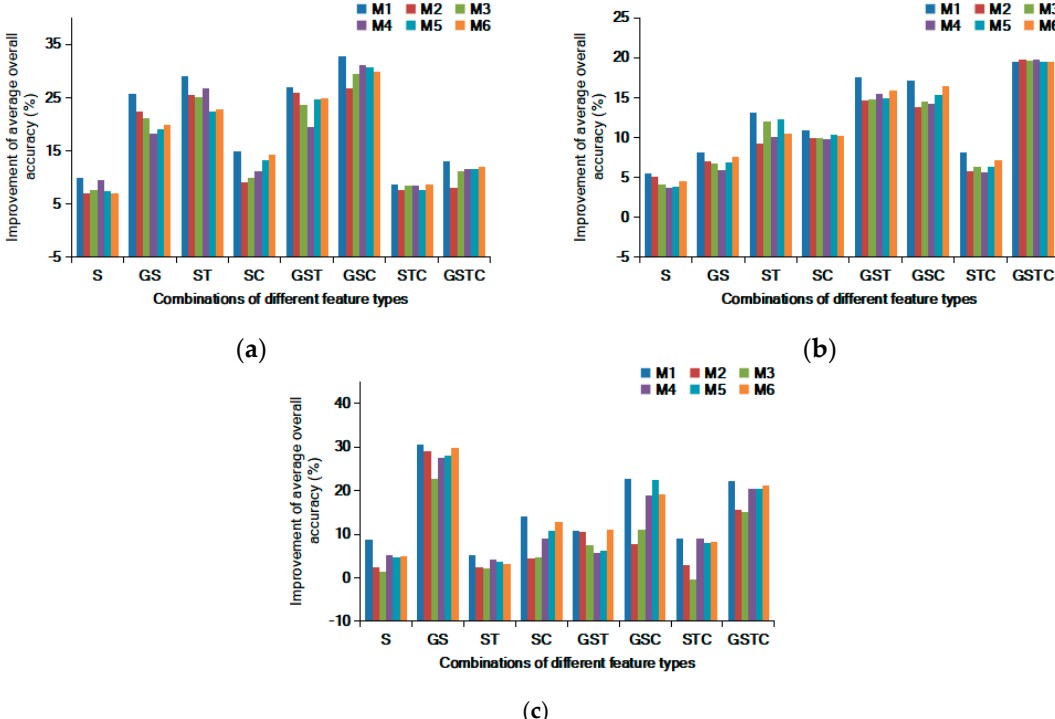

**Figure 19.** The improvement of average overall accuracies (learning rates) for the 6 AL algorithms in different feature-combination cases. (**a**) "T1;" (**b**) "T2;" and (**c**) "T3."

Aside from the average classification performance obtained by using the 20 repetitive runs, we also analyzed the highest overall accuracy that occurred in the 20 runs of different feature combination cases, as revealed in Figure 20. It is interesting to find that among the eight feature combinations, the best learning rate did not correspond to the highest overall accuracy for all of the three datasets. This demonstrates that to achieve the best classification performance by using the proposed AL method, it is still necessary to test different initial training sets, feature combinations, and numbers of samples. Such a trial-and-error strategy may be time- and labor-consuming, but the results shown in Figures 19 and 20 indicate that the proposed technique has a better chance of obtaining good classification accuracy, as compared to other AL approaches.

The patterns revealed by Figures 19 and 20 imply that the optimal object-based feature combination varies for different scenes since the best feature combinations of the three datasets are not the same. This is easy to understand because the discriminative power of a feature set tends to change according to the geo-contents of the image. Some recent studies on object-based AL have had different optimal feature combinations. Ma et al. [45] adopted three object-based feature types (shape, spectra, and texture) in their AL experiment. Their datasets included an agricultural district and two urban areas. Gray-level co-occurrence matrix (GLCM) features were incorporated into their AL strategy and were found to positively affect the classification performance. Xu et al. [44] also compared three types of features (shape, spectra, and texture) for the problem of earthquake damage mapping. They found that geometric features were very effective, and when this feature type was combined with spectral and textual information, the highest classification accuracy was obtained. The inconsistencies presented in the aforementioned studies indicate that feature selection and engineering is necessary for object-based AL methods. Additionally, it may not be optimal when all of the object-based feature types are used. To achieve the best AL performance for different images, it is suggested to test different feature configurations.

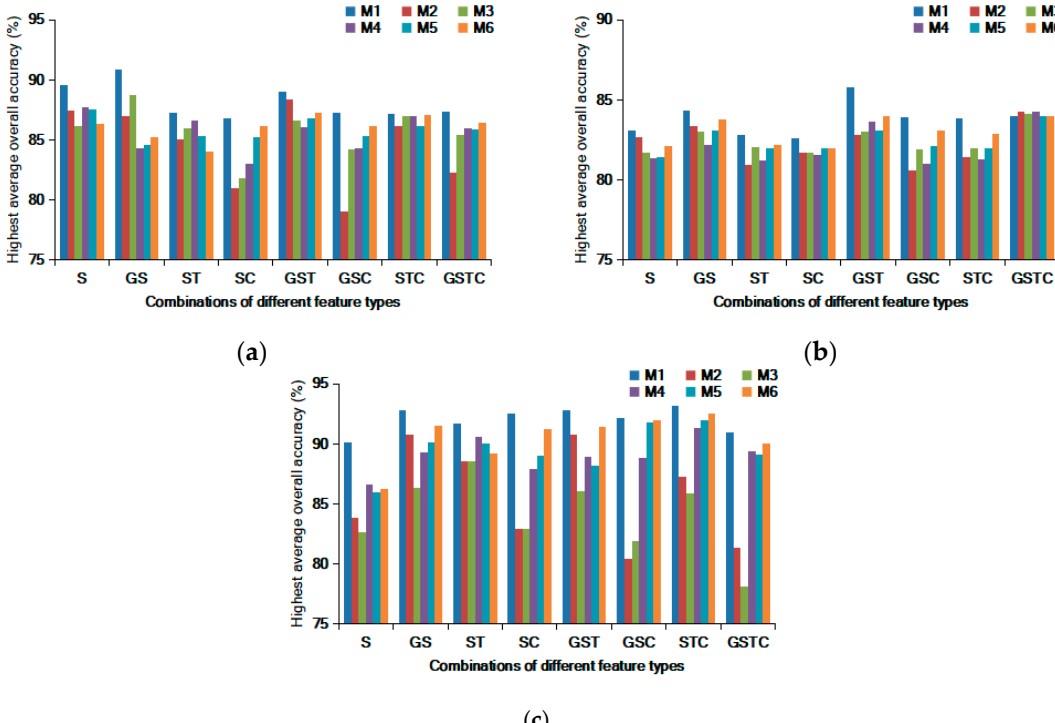

**Figure 20.** The highest overall accuracies obtained by using the 6 AL algorithms in different feature-combination cases. (**a**) "T1;" (**b**) "T2;" and (**c**) "T3.".

As for the limitation of the proposed AL technique, we have summarized two points. For one thing, according to the discussion presented in the last paragraph, feature selection should be performed by the user to achieve the optimal performance for the mapping problem at hand. This is because, for different landscape patterns, the best feature combination is not consistently the same due to the variation of the discriminative power of diverse object-based features. We recommend that users of our method select the features having good separability for their image. For example, when mapping an urban-area image, if obvious geometric differences exist for two types of buildings, shape information should be considered in the formulation of AL.

For another, there is no automatic stopping criterion for the proposed AL approach. In other words, the user has to decide when to stop the AL process. In the experiments of this article, 9 iterations are adopted to plot Figures 10–12. Though it sfigureeems that the number of iterations is sufficient for the three image pairs, it may not be adequate for other datasets. For real applications, when ground-truth validation samples are not available, it is hard for users to determine the optimal iteration number. Developing an automatic stopping method for the proposed AL algorithm is a meaningful research direction and will be our future work. However, for the current version of our AL method, the user has to set the number of iterations as the stopping criterion. We recommend employing a high value as this parameter since, in many cases, good accuracy occurs when the number of samples is high, as pointed out by Figures 16–18.

## 6. Conclusions

An object-based active learning algorithm has been proposed in this article. The objective of this method is to select the most useful segment-samples, which are to be labeled by a supervisor and then added to the training set to improve classification accuracy. In doing so, a series of binary RF classifiers and object-level features are used to quantify the appropriateness of a segment-sample. The sample with a high appropriateness value is selected with a large priority. Given that the object-based feature space is complex, it is difficult to accurately estimate sample appropriateness, but our experimental

results indicate that the proposed approach can choose effective samples, mostly thanks to the binary RF classifiers because they allow for a detailed description of the sample appropriateness by using various types of object-based features.

To validate the proposed approach, three pairs of high-resolution multi-spectral images were used. For each image pair, the first one is used for AL execution, resulting in an enlarged training set adopted for the classification task of the second image. The experimental results indicate that our AL method was the most effective in terms of improving classification accuracy, as compared to five other AL strategies. Considering that the proposed AL algorithm relies on the information of feature variables for sample selection and there are various types of object-based features, it is necessary to investigate the influences of feature combinations on the performance of an object-based AL. Thus, in our experiment, the AL-resulted classification improvements were compared for eight feature combinations, and the best combination was determined for the three datasets. It was interesting to find that the optimal feature combination varied for different datasets. This was because the discriminative power of the four feature types that were tested in this study was not the same for different landscape patterns. Accordingly, we suggest the users of our AL method test the effects of different feature combinations to achieve the best accuracy.

**Author Contributions:** T.S. proposed the idea, implemented the AL algorithm, conducted the validation experiment, and wrote the manuscript; S.Z. helped design the experiment, collected dataset, and wrote part of discussion. T.L. also helped design the experiment, and collected funding. All authors have read and agreed to the published version of the manuscript.

**Funding:** This research was jointly funded by National Key R&D Program of China, under grant number 2018YFC0406401, National Natural Science Foundation of China, under grant number 61701265, 51779116, and the Inner Mongolia Science Fund for Distinguished Young Scholars, under grant number 2019JQ06.

**Acknowledgments:** The authors want to thank the reviewers and the whole editorial team due to their helpful comments for the improvement of this paper. Also, the authors thank Inner Mongolia Autonomous Region Key Laboratory of Big Data Research and Application of Agriculture and Animal Husbandry for providing the satellite image data.

**Conflicts of Interest:** The authors declare no conflict of interest.

## Appendix A

Abbreviations used in this paper. They are provided here for the convenience of reading.

**Table A1.** Descriptions on the abbreviations and letter symbols related to an AL approach.

| Abbreviation | Description |
| --- | --- |
| AL | Active learning. |
| OBIA | Object-based image analysis. |
| RF | Random forest. |
| DT | Decision tree. |
| OAA | One-against all. |
| OAO | One against one. |
| $T$ | Training sample set. Each element of this set contains a sample and its label. |
| $C$ | Classifier. In this paper it refers to a supervised classification algorithm. |
| $U$ | Unlabeled sample set. Each element of this set only contains a sample, and its label is unknown. |
| $Q$ | Query function. It aims to measure the appropriateness of a sample in $U$. |
| $S$ | Supervisor. In most cases, especially in real operational situations, the user acts as the supervisor. $S$ aims to provide the label information for the unlabeled samples selected by a query function $Q$. |

**Table A2.** The 6 AL algorithms used in this experiment.

| Abbreviation | Description |
|:---:|:---:|
| M1 | The proposed AL technique, as delineated in Section 2. |
| M2 | An AL scheme based on entropy query metric [39]. |
| M3 | An AL approach based on breaking tie criterion. [34] |
| M4 | An AL strategy based on random sampling. |
| M5 | A multinomial logistic regression-based AL method based on a selective variance criterion [62]. |
| M6 | An object-based AL algorithm constructed by using multinomial logistic regression classifier and breaking tie metric [63,64]. |

**Table A3.** The 8 combinations of different object feature categories used in the AL experiment. The 2 defined situation aims at simplifying the analysis.

| Abbreviation | Description | 2 Defined Situations |
|:---:|:---:|:---:|
| S | Spectral features only. | Simple combination |
| GS | Geometric and spectral features. | |
| ST | Spectral and textural features. | |
| SC | Spectral and contextual features. | |
| GST | Geometric, spectral, and textural features. | Complex combination |
| GSC | Geometric, spectral, and contextual features. | |
| STC | Spectral, textural and contextual features. | |
| GSTC | The 4 types of features are all used. | |

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
