# Peer review of "Multi-Spectral Image Classification Based on an Object-Based Active Learning Approach"

_remotesensing, doi:10.3390/rs12030504_

Round 1

Reviewer 1 Report

This manuscript is very interesting. However, the writing of the manuscript seems confusion. Thus, I suggest to re-construct the manuscript,, especially for the experimental section.

Please add the AL-based methods for the extensive comparisons. Prof. Li Jun has published more related AL methods for land cover mapping.

The confusion matrix is not required for each method. But the comparisons need to be clarified.

Too many abbreviations lead the manuscript is hard to be understood. I suggest to Table to the appendix section of the manuscript.

Details of each dataset are required, including the collection date, sensor, and the location.

I am very glad to read the revised manuscript again.

Author Response

Comment 1: This manuscript is very interesting. However, the writing of the manuscript seems confusion. Thus, I suggest to re-construct the manuscript,, especially for the experimental section. Response 1: The structure of this manuscript has been re-organized to improve its quality. Details include: 1) the description of data set is put to section 3; 2) the experimental results are re-organized in the revised section 4 according to the objective of different experiments; in greater detail, section 4.1 only provide the segmentation results; the AL-related results are focused on in section 4.2; sub-section 4.2.1 describes the effects of different object feature combinations on AL performance; sub-section 4.2.2 exhibits the comparative analysis for the classification results obtained by using 6 AL approaches; lastly, sub-section 4.2.3 presents the analysis of parameter q on the performance of the proposed AL technique. Comment 2: Please add the AL-based methods for the extensive comparisons. Prof. Li Jun has published more related AL methods for land cover mapping. Response 2: 2 other competitive AL approaches are added to our experiment. The first one, represented as M5, is a recently proposed technique. It relies on a selective variance criterion to determine the unlabeled samples, and its author is one of the team led by Prof. Li Jun (62.Tan, K.; Wang, X.; Zhu, J.; Hu, J.; Li, J. A novel active learning approach for the classification of hyperspectral imagery using quasi-Newton multinomial logistic regression. International J. Remote Sens. 2018. 39, 3029-3054.). The second approach, symbolized as M6, is an object-based AL approach firstly proposed by Prof. Li Jun (63.Li, J.; Bioucas-Dias, J.; Plaza, A. Semisupervised hyperspectral image segmentation using multinomial logistic regression with active learning. IEEE Trans. Geosci. Remote Sens. 2010. 48, 4085-4098..). We used a relatively recnet variant of this approach, which is detailed by Guo et al (Guo, J.; Zhou, X.; Li, J.; Plaza, A.; Prasad, S. Superpixel-based active learning and online feature importance learning for hyperspectral image analysis. IEEE J. Selec. Topic. Appl. Earth Observ. Remote Sens. 2016. 10, 347-359). Comment 3: The confusion matrix is not required for each method. But the comparisons need to be clarified. Response 3: The confusion matrices have been removed, but the class-specific accuracy values are preserved, as can be seen in Table 10, 11, 12 in sub-section 4.2.2 of the revised manuscript. The related comparisons are clarified, as can be seen in the revised sub-section 4.2.2. Comment 4: Too many abbreviations lead the manuscript is hard to be understood. I suggest to Table to the appendix section of the manuscript. Response 4: the abbreviations are all put to the appendix section, as can be found in the 3 tables in Appendix. Comment 5: Details of each dataset are required, including the collection date, sensor, and the location. Response 5: the descriptions on the acquisition date and location of the datasets have been detailed from the third to the fifth paragraphs in sub-section 3.1. The sensor information is added to the second paragraph of sub-section 3.1.

Reviewer 2 Report

The paper is much improved in this revised version. I believe that the current form is acceptable for publication.

Author Response

Since there is no comments in this round of review, no response is needed here.

The authors of this manuscript sincerely thank this reviwer!

Round 2

Reviewer 1 Report

Thank you very much for your response and revision. Now the manuscript seems better. I have no comments about the content of the manuscript.
However, to guarantee the quality of your manuscript, the resolution of Figures 6, 1o, 11 and 12 is suggested to be improved. Excel is not a good choice for drawing figures. The style of these figures should be consistent with that of Figure 16. 

Author Response

The figures (Figure 6, 10, 11, and 12) have been remade to enhance their quality, so that their styles are consistent with that of Figure 16. Details can be found in the revised Figure 6, 10, 11, and 12.

This manuscript is a resubmission of an earlier submission. The following is a list of the peer review reports and author responses from that submission.

Round 1

Reviewer 1 Report

Please see my comments in the attached PDF.

Author Response

Point-by-point responses are provided below. For the convenience of further reviewing, all of the revised parts are highlighted by red colour in the uploaded manuscript.

Comment 1: In the abstract, the problem in the current research should be clarified. In addition, what problem you will solve in your proposed approach should be given. Furthermore, the advantage or the steps of your proposed approach should be presented. The quantitative compassion between your proposed approach and the state-of-the-art methods should be provided.

Response 1: The abstract has been almost re-written, so that the problem, advantage, and the quantitative results can be clearly demonstrated. Details can be found in the revised manuscript.

Comment 2: You had better present the quantitatively improvement in terms of the comparisons between the proposed approach and the state-of-the-art methods.

Response 2: The quantitative improvement of the proposed method as compared to the other methods are clearly provided at the end of abstract.

Comment 3: The literature reviews are not sufficient. More new literature should be reviewed. Such as the literatures as followings:

ï‚· 1 Multi-Scale Object Histogram Distance for LCCD Using Bi-Temporal Very-High-Resolution Remote Sensing Images

ï‚· 2 Novel Object-Based Filter for Improving Land-Cover Classification of aerial with very high spatial resolution

ï‚· 3 Automatic Object-oriented, Spectral-spatial Feature Extraction Driven by Tobler’s First Law of Geography for Very High-Resolution Aerial Imagery Classification

ï‚· 4 Novel Adaptive Histogram Trend Similarity Approach for Land Cover Change Detection by Using Bitemporal Very-High-Resolution Remote Sensing Images

ï‚· 5 Salient Band Selection for Hyperspectral Image Classification via Manifold Ranking,”TNNLS.

ï‚· 6 “Hyperspectral Image Classification via Multi-Task Joint Sparse Representation and Stepwise MRF Optimization,”TCYB.

ï‚· 7 “Hierarchical Feature Selection for Random Projection,”TNNLS.

ï‚· 8 Cheng Shi, Chiman Pun. “Adaptive Multi-scale Deep Neural Networks with Perceptual Loss for Panchromatic and Multispectral Images Classification”, Information Sciences, vol. 490, pp. 1-17, 2019

Response 3: All of the articles listed above have been considered and added to the literature review of the revised introduction.

Comment 4: In the experiment section, the label of Figure 4 should be added, if each node has the label, the readability of the figure will be improved.

Response 4: The class labels of the selected objects are shown in Figure 5(c) and (f), as can be observed in the revised Figure 5.

Comment 5: All the text in the table should be centered.

Response 5: The tables are edited according to this comment.

Comment 6: T1 and T2 should be updated by “T1” and “T2”

Response 6: The words T1 and T2 in the manuscript are all updated as "T1" and "T2".

Comment 7: “Simplified Name” should be replaced by “abbreviation”, as shown in Table 7,8.

Response 7: The words " Simplified Name " in Table 7 and 8 are corrected as “abbreviation”.

Comment 8: The conclusion should be rewritten, because it seems incredible and it is unclearly.

Response 8: The conclusion is completely re-written, and details can be seen in the revised manuscript.

Reviewer 2 Report

The paper is generally well written but needs major language editing. However, I have several major issues with the current submission:

The paper needs major editing with respect to correct usage of the English language. The motivation is not convincing. While it is true that RS is often used when areas are difficult and/or dangerous to access, AL can barely help in this case. If for those reasons only a limited training set is available, AL is able to tell which other parts of the data would help most of labels for them would be available. However, if the corresponding person is not on site anymore, this helps nothing - as it is impossible to go there again. The only scenario where it might be helpful is, when the corresponding AL procedure is carried out on site. In this scenario, however, a lot of other factors would need to be considers (e.g. run time, whether the software can run on portable devices, etc.). Also, the cost of acquiring those new samples on site (e.g. time spent on the way, accessibility etc.) would need to be considered. The common scenario of AL is, however, not using on site, but offline after data acquisition in order to limit the label cost by manually analysing the data. AL aims to prevent that human experts spend hours labelling data that is of little use for the classifier and instead spent minutes on labelling only the important parts. This is only possible if reference data can be acquired in this way (e.g. off site manual inspection of the data). The article is highly misleading about this distinction. he presented methodology is quite standard. The only change to the standard procedure is - as far as I can see - the usage of a RF. This barely qualifies as a novel contribution. Thus, the paper either lacks novelty or fails to clearly state what is novel. The evaluation is flawed. First of all, if the paper aims to introduce a novel AL method and is not a case study, standard (public) datasets should be used instead of ones own data. Only then the experiments have a chance to being repeatable and comparable by others. Second, a single image is used for training and testing. This is in itself already a bad thing as it leads to biased error estimates. However, the applied train and test procedure make this problem much worse. Assume the data contain samples that a) are correctly labelled with high confidence, b) correctly labelled with low confidence, c) are wrongly labelled with high confidence, d) wrongly labelled with low confidence. Obviously, only the samples in case c and d contribute to the first estimate of the test error (since they are wrongly classified). The proposed AL scheme now selects samples from case b and d and adds them to the training set. Consequently, less wrongly labelled samples remain in the test set (only samples in case c). This means, that the estimated test error gets better because more and more wrongly estimated samples are removed from the test set - but not necessarily because the classifier does really improve. For an unbiased estimate, the test set should stay constant for all AL iterations. The paper indicates at multiple points that hyperparameters of the proposed approach had been optimized. Since only one image is used, that means they have been optimized on this image. As stated above, the proposed method only makes sense if the data can be manually labelled offline. In this case, the question is what is faster / more efficient: Running the proposed AL approach including optimizing all hyperparameters; OR simply manually labelling the whole image.

Author Response

Point-by-point responses are provided below. For the convenience of further reviewing, all of the revised parts are highlighted by red colour in the uploaded manuscript.

Comment 1: The paper needs major editing with respect to correct usage of the English language.

Response 1: The English Language has been significantly revised, including removing inappropriate expression and correcting erroneous grammar. Details can be found in the red-color highlighted texts of the revised manuscript.

Comment 2: The motivation is not convincing. While it is true that RS is often used when areas are difficult and/or dangerous to access, AL can barely help in this case. If for those reasons only a limited training set is available, AL is able to tell which other parts of the data would help most of labels for them would be available. However, if the corresponding person is not on site anymore, this helps nothing - as it is impossible to go there again. The only scenario where it might be helpful is, when the corresponding AL procedure is carried out on site. In this scenario, however, a lot of other factors would need to be considers (e.g. run time, whether the software can run on portable devices, etc.). Also, the cost of acquiring those new samples on site (e.g. time spent on the way, accessibility etc.) would need to be considered. The common scenario of AL is, however, not using on site, but offline after data acquisition in order to limit the label cost by manually analysing the data. AL aims to prevent that human experts spend hours labelling data that is of little use for the classifier and instead spent minutes on labelling only the important parts. This is only possible if reference data can be acquired in this way (e.g. off site manual inspection of the data). The article is highly misleading about this distinction. he presented methodology is quite standard. The only change to the standard procedure is - as far as I can see - the usage of a RF. This barely qualifies as a novel contribution. Thus, the paper either lacks novelty or fails to clearly state what is novel.

Response 2: First, we thank reviewer 2 for his constructive comment, which helps us realize this flaw in the motivation. The analysis of this comment is very deep and informative, making us understand what is the major problem in the description of the motivation. Accordingly, the motivation part has been completely rewritten, so that the meaning of this study can be presented in a clearer way. Also, the texts introducing the innovation of this article are revised, in order to better demonstrate the novelty of this paper. Related details can be seen in the revised introduction, and some explanations are provided as follows.

Regarding the motivation, we agree with the reviewer about his statement "AL can barely help when areas are difficult or dangerous to access." We admit that AL cannot solve the problem that the information of some samples cannot be acquired. In the original version of this article, the motivation left a wrong impression that the proposed AL approach aimed to solve the problem of a small sample set due to inaccessibility to the study area, which is indeed misleading and inappropriate. We think the true meaning of AL is to lower the cost of sample collection by guiding the user to select the most effective samples, so that redundant and/or insignificant samples can be avoided, and accordingly classifier performance can be improved. Based on this, what I want to express is that the cost of collecting samples is generally high, and one of the reasons is limited accessibility due to danger or inconvenience. Thus, in many real applications, the key problem is how to obtain a sufficiently high classification accuracy with a limited number of training samples. AL aims to provide a solution to this issue, and this is the most significant meaning of AL.

Aside from the meaning of AL, another aspect of the motivation of this research resides in the application of object-based image analysis (OBIA). As is well known and frequently reported by many literature, OBIA becomes increasingly popular in remote sensing community during recent years, but studies on the combination of OBIA and AL are relatively rare. To the best of our knowledge, only a small number of papers have addressed this research direction. Additionally, most studies applying AL to remote sensing image classification are within the scope of pixel-based analysis, which is quite different from OBIA, since the two paradigms use different processing unit. Pixel-based methods only consider pixel, and thus pixel-based AL can only select pixel as samples. On the contrary, when it comes to OBIA, segment or region becomes the processing unit. Although the change of processing unit may seem insignificant, a major issue arises, which is mainly due to the change of feature space. In pixel-based analysis, features are calculated based on a single pixel or a window of pixels centered at the target pixel. In this context, only the information of a limited (or regulated) spatial range can be captured. However, the feature space of OBIA is much more complicated, since the features are derived by using the pixels belonging to a segment. Thus, more information such as geometric and adaptive spatial contextual features can be utilized in OBIA. The more complicated feature space brings about a new challenge to the traditional AL algorithms, because AL relies on the information of feature space to quantify uncertainty and reflect informativeness. Accordingly, the more complicated feature set in OBIA requires more appropriate AL methods, which motivates this work.

The innovation of this study has two aspects. First, one-against-one (OAO) random forest (RF) model is designed to quantify uncertainty. Although RF has been applied to AL and OBIA, one-against-one (OAO) model has not been utilized. Since OAO provides detailed description for feature space, we believe that this strategy can help increase the performance of AL in the context of OBIA, since the feature space of OBIA is generally much more complicated than the counterpart of pixel-based image analysis. Second, in the framework of OBIA, 4 different types of features (geometric, spectral, textural, and contextual features) are used in the evaluation experiment. Although all of these features have been investigated in previous OBIA studies, none of them investigated their effects on AL performance. This article presents the influences of various combinations of the 4 feature types on AL performance, which can be deemed as a novel contribution in terms of experimental design.

Comment 3: The evaluation is flawed. First of all, if the paper aims to introduce a novel AL method and is not a case study, standard (public) datasets should be used instead of one's own data. Only then the experiments have a chance to being repeatable and comparable by others. Second, a single image is used for training and testing. This is in itself already a bad thing as it leads to biased error estimates. However, the applied train and test procedure make this problem much worse. Assume the data contain samples that a) are correctly labelled with high confidence, b) correctly labelled with low confidence, c) are wrongly labelled with high confidence, d) wrongly labelled with low confidence. Obviously, only the samples in case c and d contribute to the first estimate of the test error (since they are wrongly classified). The proposed AL scheme now selects samples from case b and d and adds them to the training set. Consequently, less wrongly labelled samples remain in the test set (only samples in case c). This means, that the estimated test error gets better because more and more wrongly estimated samples are removed from the test set - but not necessarily because the classifier does really improve. For an unbiased estimate, the test set should stay constant for all AL iterations. The paper indicates at multiple points that hyperparameters of the proposed approach had been optimized. Since only one image is used, that means they have been optimized on this image. As stated above, the proposed method only makes sense if the data can be manually labelled offline. In this case, the question is what is faster / more efficient: Running the proposed AL approach including optimizing all hyperparameters; OR simply manually labelling the whole image.

Response 3: This reviewer is sincerely thanked again for this piece of comment, since the analysis is quite detailed and deep. However, we are afraid that there is an error in the analysis, which is probably due to a misunderstanding to our approach and experiment. The error is in the explanation of the second point. The reviewer holds that the performance of our algorithm gets better because more wrongly estimated samples are removed from the test set. This is incorrect, since there is no wrongly estimated samples during the whole procedure. The proposed AL method only estimates uncertainty, and AL does not output a label prediction for the test samples, let alone whether their estimated labels are correct or not. In addition, the overall accuracy values are all derived by using the validation set, which remains constant throughout the experiment. The validation set is illustrated in Figure 3(b) and (d). Consequently, it can be seen that the classification accuracy values are not biased. We guess that this misunderstanding is probably attributed to the name of "test set". The meaning of "test" may make him think that the set is used for accuracy evaluation, but this is not true. In fact, the test set is used for the appropriateness/uncertainty measuring process of AL, and each element in the test set is tested by the AL procedure, which outputs an uncertainty value for the tested element. As mentioned earlier in Response 2, the objective of AL is to select the most effective samples (note that AL can only improve the performance of a classifier by providing more effective samples, so the raise of overall accuracy as shown in Figure 6, 8, 10, 11, 13, 15 reflects that better samples are identified by using the proposed AL), and the selected samples are labeled by the supervisor, who is generally the user himself. Thus, the elements in the test set do not contain label information, and they are not used for accuracy computation.

As for the first point of this comment, we are afraid that the two images have to be adopted in this study, since the two acknowledged projects supporting this work require us to do so. What's more, as mentioned in the last paragraph, the evaluation is not biased, thus we hold that there is no need to use other images for further evaluation. Besides, almost all of the previous AL-related studies have used the training and evaluating samples extracted from only one image to evaluate AL method. Such as the examples: (just to list a few)

Li, J.; Bioucas-Dias, J. M.; Plaza, A. Hyperspectral Image Segmentation Using a New Bayesian Approach With Active Learning. IEEE Trans. Geosci. Remote Sensing 2011. 49, 3947-3960. Chenying Liu et al. Superpixel-Based Semisupervised Active Learning for Hyperspectral Image Classification. IEEE JOURNAL OF SELECTED TOPICS IN APPLIED EARTH OBSERVATIONS AND REMOTE SENSING. 2018. Saeid Niazmardi, Saeid Homayouni & Abdolreza Safari. A computationally efficient multi-domain active learning method for crop mapping using satellite image time-series. International Journal of Remote Sensing. 2019. Xu, Z.; Wu, L.; Zhang, Z. Use of active learning for earthquake damage mapping from UAV photogrammetric point clouds. International J. Remote Sensing 2018. 39, 5568-5595.

But we do agree with the reviewer about making this experiment repeatable, thus, the two datasets have been made freely available online through the following link:

https://pan.baidu.com/s/1Gpvoj06b5iITxrNDeCua4g

Reviewer 3 Report

There are comments on the manuscript that the authors should be taken into account in the next revision.

The motivation for the utilization of active learning in this field is not clear. What kinds of the contribution of active learning to address current problems in the remote sensing image classification/segmentation? The proposed algorithm is not presented understandably. For example, in Fig. 2, the arrows of processing flows should be numbered to clarify the process ordering. It looks the performance/accuracy of the proposed active learning mostly depends on the feature extraction. But the performance analysis are not mentioned in experiments. The proposed method is investigated under different parameter configuration, but no method comparison is given (with other state-of-the-art approaches, not only the variations of active learning)?

Author Response

Point-by-point responses are provided below. For the convenience of further reviewing, all of the revised parts are highlighted by red colour in the uploaded manuscript.

Comment 1: The motivation for the utilization of active learning in this field is not clear. What kinds of the contribution of active learning to address current problems in the remote sensing image classification/segmentation?

Response 1: The motivation part has been completely rewritten, and details can be seen in the last but 3 paragraph of the revised introduction. Some explanations are given as follows. The main idea is that, we have found that there are very few efforts dedicated to the development of object-based AL approaches. To the best of our knowledge, most AL-related studies in remote sensing are within the scope of pixel-based image analysis, which is quite different from object-based image analysis (OBIA). The major difference is feature space. Pixel-based features are generally less complicated than the counterparts of object-based ones, since geometric, adaptive spatial contextual information can be utilized in OBIA, while they cannot be adopted in pixel-based paradigm. What's more, AL relies on information of feature variables to quantify the informativeness of samples, in order to select the most informative/effective ones. Therefore, the more complicated feature space of OBIA requires more advanced AL algorithms. This primarily motivates the work of this paper.

Comment 2: The proposed algorithm is not presented understandably. For example, in Fig. 2, the arrows of processing flows should be numbered to clarify the process ordering. It looks the performance/accuracy of the proposed active learning mostly depends on the feature extraction. But the performance analysis are not mentioned in experiments.

Response 2: More details of the proposed algorithm have been provided, which can be seen in section 2.4 of the revised manuscript.

The arrows in Figure 2 are all numbered, so that the order of the whole process can be illustrated. Details can be seen in the revised Figure 2.

As for the evaluation experiment, the proposed algorithm has been thoroughly validated, which is explained in detail in the following. As described in the motivation, this paper aims to propose an object-based AL method. The objective of AL is to identify the most effective/informative samples to be labeled by the user, and then the selected samples are added to the training set, so that classification performance can be raised in an optimal manner. Under this circumstance, AL methods are generally evaluated by testing whether their selected samples can lead to improved classification accuracy. Such experiments are frequently reported in previous literature, such as (just to list a few examples):

Ma, L.; Fu, T.; Li, M. Active learning for object-based image classification using predefined training objects. International J. Remote Sensing 2018. 39, 2746-2765. Sun, S.; Zhong, P.; Xiao, H.; Wang, R. Active learning with gaussian process classifier for hyperspectral image classification. IEEE Trans. Geosci. Remote Sensing 2015. 53, 1746-1760 Huo, L.; Tang, P. A batch-mode active learning algorithm using region-partitioning diversity for SVM classifier. IEEE J. Selected Topic. Applied Earth Observ. Remote Sensing 2014. 7, 1036-1046.

In the experimental section of this article, the proposed AL method has been compared to other standard AL approaches, as introduced in Table 7. The comparative results are plotted and shown in Figure 6, 8, 11, 13. The comparison is thorough, because different scenarios of feature combinations are tested. The reason of this statement is: the target problem of this work is to apply AL to OBIA (note that most previous literature only use AL in pixel-based analysis), and the most significant difference between object-based and pixel-based image analysis is feature space, since the former can use more feature types and more complicated feature variables, such as geometric and adaptive spatial contextual features. Although many OBIA-related studies have analyzed the effects of different types of object-based features on classification performance, almost none of them have investigated the influences of different object-based feature sets on AL. Thus, it is highly necessary to test what are the effects of feature combinations on object-based AL performance.

To sum up, there is a comparison experiment in this paper, but the experiment has been conducted in different feature combination scenarios. Thus, the experiment of this paper is thorough.

Comment 3: The proposed method is investigated under different parameter configuration, but no method comparison is given (with other state-of-the-art approaches, not only the variations of active learning)?

Response 3: We think that it is of little meaning to compare the proposed method to other non-AL state-of-the-art approaches, since their objectives are different. AL aims to find the most effective samples for classifier training, so that the best classification performance of the classifier can be achieved (rather than further enhance classifier performance). However, the objective of the non-AL state-of-the-art classification methods is to improve classification performance by improving separability of the classifier model or by enhancing the separability of the feature set used for classification.

Accordingly, it is clear that the direct objective of AL studies is to identify the most useful samples for training, and improving classification performance is secondary and indirect. On the contrary, other state-of-the-art algorithms aim to straightly improve classification performance. This difference in the research objective makes it meaningless to compare an AL method and a non-AL classification scheme.

Besides, as explained in Response 2, the experiment has already demonstrated the comparative analysis and validation.

Reviewer 4 Report

I have serious concerns on the presentation of this paper. The significance of this paper is quite high but its presentation unfortunately hides it. In my case I had to read it because I had to review it but you will surely loose a young researcher.

First lets start from the title:

Object-based active learning for high resolution remote sensing image classification

It should be

Multi-spectral  image classification based on an object oriented active learning approach.

Please transfer one of the samples in the experiment and based on this describe

1) Segmentation. How you do the segmentation and how it is used in active learning. Describe your whole methodology based on the scene. Segmentation, active learning, random forests etc. 

Like this I can't accept it. This paper needs to follow this road. 

Kindly do this and I will be willing to accept the paper with minor/major corrections. 

The ideas are really nice but the presentation can't pass them.

Author Response

Point-by-point responses are provided below. For the convenience of further reviewing, all of the revised parts are highlighted by red colour in the uploaded manuscript.

Comment 1: I have serious concerns on the presentation of this paper. The significance of this paper is quite high but its presentation unfortunately hides it. In my case I had to read it because I had to review it but you will surely loose a young researcher. First let's start from the title:

Object-based active learning for high resolution remote sensing image classification

It should be

Multi-spectral  image classification based on an object oriented active learning approach.

Response 1: The title of this paper has been changed as "Multi-spectral image classification based on an object-based active learning approach".

Note that the term "object-based" is used instead of "object-oriented", because we think that the former is more commonly used in object-based image analysis (OBIA) community (as discussed in a previous frequently-cited paper: Blaschke, T., 2010. Object-based image analysis for remote sensing. ISPRS J. photogramm. Remote Sens. 65, 2-16.). And researchers in this community are target readers for this article.

Comment 2: Please transfer one of the samples in the experiment and based on this describe 1) Segmentation. How you do the segmentation and how it is used in active learning. Describe your whole methodology based on the scene. Segmentation, active learning, random forests etc.

Like this I can't accept it. This paper needs to follow this road.

Kindly do this and I will be willing to accept the paper with minor/major corrections.

The ideas are really nice but the presentation can't pass them.

Response 2: The whole working flow of the proposed AL method, together with the details of how the AL performs in an object-based image classification, are described in sub-section 2.4 of the revised manuscript. Explanations pertaining to segmentation and random forest classification are presented in detail in sub-section 2.4.2, while the delineation of the proposed AL is focused on in sub-section 2.4.1. Figure 2 is re-made to better demonstrate the whole process and the details of the proposed AL methodology.

Note that since the image segmentation step of this study is completely unsupervised, and the samples are only used in the steps after segmentation, thus no samples are needed for image segmentation.

Round 2

Reviewer 1 Report

Thank you very much for your response.

All the concerned problems have been well solved. However, when I read the manuscript again, it was found that the language of the manuscript requires extensive polishing. So, I suggest checking the grammar by an English native speaker.

In fact, language editing of the MDPI is a good choice.

Reviewer 4 Report

This paper describes an active learning approach to train a classifier (Random Forests) using as features objects instead of pixels. Please bear in mind that the feature space is a d-dimensional space as normally is expected in a classifier. They choose in their active learning to label with high uncertainty and then retrain the Random Forest expecting it to generalize better. As we say the more data the better...

Please kindly describe briefly how M2, M3 are getting computed and their difference with your model. Like this the reader can't appreciate your work.